# Quantitative neuroanatomy for connectomics in *Drosophila*

Casey M Schneider-Mizell[1†], Stephan Gerhard[1,2,3†], Mark Longair[2,3],
Tom Kazimiers[1], Feng Li[1], Maarten F Zwart[1], Andrew Champion[1],
Frank M Midgley[1], Richard D Fetter[1], Stephan Saalfeld[1], Albert Cardona[1*]

[1]Janelia Research Campus, Howard Hughes Medical Institute, Ashburn, United States; [2]Institute of Neuroinformatics, University of Zurich, Zürich, Switzerland; [3]Eidgenössische Technische Hochschule Zürich, Zurich, Switzerland

*For correspondence: cardonaa@janelia.hhmi.org

†These authors contributed equally to this work

Competing interests: The authors declare that no competing interests exist.

**Abstract** Neuronal circuit mapping using electron microscopy demands laborious proofreading or reconciliation of multiple independent reconstructions. Here, we describe new methods to apply quantitative arbor and network context to iteratively proofread and reconstruct circuits and create anatomically enriched wiring diagrams. We measured the morphological underpinnings of connectivity in new and existing reconstructions of *Drosophila* sensorimotor (larva) and visual (adult) systems. Synaptic inputs were preferentially located on numerous small, microtubule-free 'twigs' which branch off a single microtubule-containing 'backbone'. Omission of individual twigs accounted for 96% of errors. However, the synapses of highly connected neurons were distributed across multiple twigs. Thus, the robustness of a strong connection to detailed twig anatomy was associated with robustness to reconstruction error. By comparing iterative reconstruction to the consensus of multiple reconstructions, we show that our method overcomes the need for redundant effort through the discovery and application of relationships between cellular neuroanatomy and synaptic connectivity.

## Introduction

Mapping neuronal circuits from electron microscopy (EM) volumes is hard (*Helmstaedter, 2013*). Manually working through large volumes is slow and prone to attentional errors (*Kreshuk et al., 2011*; *Helmstaedter et al., 2011*). Combining multiple independent reconstructions of the same neuron can reduce errors (*Helmstaedter et al., 2011*; *Kim et al., 2014*) at the cost of multiplying the required labor. Current computational approaches operate only with 'local' information, that is, the EM micrographs and algorithmically detected fine structures such as cell membranes and mitochondria. They are therefore sensitive to noise (*Jain et al., 2010*), particularly in anisotropic EM data where the smallest neurites may be thinner than the thickness of individual serial sections (*Veeraraghavan et al., 2010*; *Helmstaedter, 2013*). Machine-generated neuron reconstructions are therefore proof-read by humans (*Chklovskii et al., 2010*; *Haehn et al., 2014*).

Expert neuroanatomists are able to resolve ambiguities that novices and current algorithmic approaches cannot by using large-scale features of neurons to inform decisions made at the level of nanometer-scale image data. For example in *Drosophila*, where neurons are highly stereotyped, large branches in an EM reconstruction of a given cell can be confirmed by comparing the observed anatomy to that of homologous cells from light microscopy data or other reconstructions (*Takemura et al., 2013*; *Ohyama et al., 2015*). This suggests that one way to improve the toolkit for neuron reconstruction and circuit mapping is to facilitate the use of cell- and circuit-level features to find and resolve errors and ambiguities.

**eLife digest** The nervous system contains cells called neurons, which connect to each other to form circuits that send and process information. Each neuron receives and transmits signals to other neurons via very small junctions called synapses. Neurons are shaped a bit like trees, and most input synapses are located in the tiniest branches. Understanding the architecture of a neuron's branches is important to understand the role that a particular neuron plays in processing information. Therefore, neuroscientists strive to reconstruct the architecture of these branches and how they connect to one another using imaging techniques.

One imaging technique known as serial electron microscopy generates highly detailed images of neural circuits. However, reconstructing neural circuits from such images is notoriously time consuming and error prone. These errors could result in the reconstructed circuit being very different than the real-life circuit. For example, an error that leads to missing out a large branch could result in researchers failing to notice many important connections in the circuit. On the other hand, some errors may not matter much because the neurons share other synapses that are included in the reconstruction.

To understand what effect errors have on the reconstructed circuits, neuroscientists need to have a more detailed understanding of the relationship between the shape of a neuron, its synaptic connections to other neurons, and where errors commonly occur. Here, Schneider-Mizell, Gerhard et al. study this relationship in detail and then devise a faster reconstruction method that uses the shape and other properties of neurons without sacrificing accuracy. The method includes a way to include data from the shape of neurons in the circuit wiring diagrams, revealing circuit patterns that would otherwise go unnoticed.

The experiments use serial electron microscopy images of neurons from fruit flies and show that, from the tiniest larva to the adult fly, neurons form synapses with each other in a similar way. Most errors in the reconstruction only affect the tips of the smallest branches, which generally only host a single synapse. Such omissions do not have a big effect on the reconstructed circuit because strongly connected neurons make multiple synapses onto each other.

Schneider-Mizell, Gerhard et al.'s approach will help researchers to reconstruct neural circuits and analyze them more effectively than was possible before. The algorithms and tools developed in this study are available in an open source software package so that they can be used by other researchers in the future.

Crucially, different errors do not alter the wiring diagram equally. Missing small dendrites can be acceptable. Useful and reproducible wiring diagrams can be created even when omitting 56% of all postsynaptic sites (*Takemura et al., 2013*), but missing a single large branch hosting all the synapses in one neuropil region could omit connectivity to entire populations of partners. Prioritizing proof-reading time toward those errors that most significantly affect the interpretation of the data improves reconstruction efficiency (*Plaza et al., 2012*; *Kim et al., 2014*).

To understand the effect of reconstruction errors on measured synaptic connectivity, we need to understand the relationship between synaptic connectivity and cellular neuroanatomy. Mesoscale anatomy, particularly the placement of large branches, is a key component of circuit structure (*Zlatić et al., 2003*, *2009*; *Wu et al., 2011*; *Couton et al., 2015*). Similarly, the connectivity graph of a stereotyped circuit can relate back to anatomy by consideration of the location of the synaptic sites between pairs of neurons. However, little is known about the smallest scales of synaptic connectivity, the distribution of individual synapses on a neuron. Microtubule-free and actin-rich structures have been identified as key sites of excitatory input in the adult *Drosophila* visual system (*Scott et al., 2003*; *Leiss et al., 2009*), but it is unclear how ubiquitous these are in the nervous system.

Here, we describe a collection of quantitative anatomical and connectivity features across scales, from fine dendritic branches to multi-neuron graphs, and tools for measuring them to swiftly and accurately map a wiring diagram from EM. We implemented the calculation and visualization of such features on-demand as an extension of the web-based large image data viewer CATMAID

(*Saalfeld et al., 2009*). We propose a novel method for interactively using these features to reconstruct neuronal circuits through iterative proofreading at the level of both EM images and higher level features. We validated this approach by comparing the speed and accuracy of our iterative method to a consensus method, where multiple independent reconstructions are used to calculate regions of agreement across individuals (*Helmstaedter et al., 2013*). Because the detection of high-impact errors can occur concurrently with reconstruction via interactive analysis, our tool removes the need for time-consuming repeated reconstructions (*Helmstaedter et al., 2013*; *Kim et al., 2014*). Moreover, because reconstructed neurons did not need to be hidden to ensure independence between repeated reconstructions, our method facilitates concurrent, synergistic collaboration between expert neuroanatomists who, for example, map circuits in different brain regions that happen to spatially overlap or synaptically interact. We demonstrate our methods by mapping a sensorimotor circuit in the *Drosophila* larva from proprioceptive sensory neurons to motor neurons.

## Results

### Collaborative circuit mapping

We extended the web-based image data viewer CATMAID (*Saalfeld et al., 2009*) to enable a geographically distributed group of researchers to map neuronal circuitry. A neuron is reconstructed with a skeleton, a directed tree graph with one or more nodes in every cross-section of neurite in an EM volume (*Helmstaedter et al., 2011*; *Cardona et al., 2012*). Nodes have a spatial coordinate, as well as metadata including authorship, timestamp, review status, and optional annotations such as a radius value, text labels. Importantly, nodes also have a confidence value that can be lowered to indicate uncertainty in following a branch. Where possible, we root skeletons at the soma to model the anatomical notions of proximal and distal in the data structure.

Synapses (*Figure 1A* and *Figure 1—figure supplement 1*) are annotated as a relation from a node on the presynaptic neuron skeleton to an intermediate 'connector node' and then to a node of a postsynaptic neuron skeleton. To express the polyadic nature of insect synapses (*Meinertzhagen and O'Neil, 1991*), connector nodes can have multiple postsynaptic 'targets', but only one presynaptic 'source'. Reconstructions are immediately synchronized across all collaborators to avoid duplicate or conflicting work, and to take advantage of existing reconstructions to aid further reconstruction and circuit discovery.

As a case study of our method, we focused on sensorimotor circuits in an abdominal segment of the first instar *Drosophila* larval central nervous system (CNS) using an EM volume covering one and a half abdominal segments (*Ohyama et al., 2015*). In total for this work, nine lab members reconstructed and proofread 425 neuronal arbors spanning 51.8 mm of cable, with 24,068 presynaptic and 50,927 postsynaptic relations, (see 'Materials and methods' for details). Reconstruction time was 469 hours for reconstruction with synapse annotations plus 240 hours for review (see below), for an average rate of ~73 microns of proofread arbor with synapses per hour.

### Microtubule-free twigs are the principal site of synaptic input

To be able to use neuronal anatomy to guide circuit reconstruction, it was crucial to better understand the distribution of synaptic input onto *Drosophila* neurons. We started by looking in detail at the relationship between the synaptic inputs (*Figure 1A–B*) and microtubule cytoskeleton (*Figure 1C–E*) in EM reconstructions of neurons from different regions of the nervous system and life stages. For a diverse collection of neurons, we marked all locations where the arbor continued distal to a microtubule-containing process (*Figure 1E*, *Figure 2A*). We call such a terminal branch a 'twig'. By definition, all twigs have their base on a microtubule-containing backbone shaft. Following the classification in Leiss et al. (*Leiss et al., 2009*), a spine is a twig with a maximal depth of less than 3 μm and that is not a presynaptic varicosity (*Figure 2A*).

We found twigs in all neurons investigated, across multiple CNS regions and life stages of *Drosophila*, and in all cases, they were the dominant sites of synaptic input (*Figure 2B–F*). We first considered larval motor neurons aCC and RP2 (*Landgraf et al., 1997*), which have functional and structural similarities to vertebrate neurons (*Sánchez-Soriano et al., 2005*; *Nicolï et al., 2010*; *Günay et al., 2015*). In the first instar CNS, we find aCC and RP2 have numerous twigs, which together host more than 80% of their total number of postsynaptic sites (*Figure 2B*). We found a

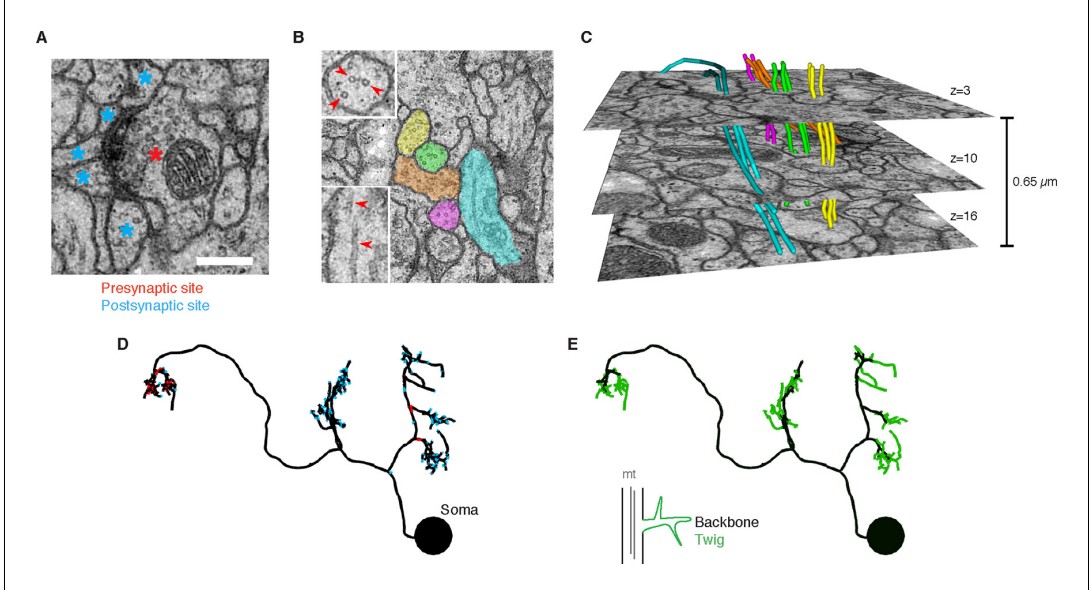

**Figure 1.** EM ultrastructure shows synapses and microtubule cytoskeleton. (**A**) EM micrograph of a typical *Drosophila* synapse with a single presynaptic site (red asterisk) and multiple postsynaptic sites (blue asterisks). Scale bar is 200 nm. (**B**) Microtubules in neural processes are visible in EM sections whether cut transverse (top inset, red arrowheads) or obliquely (bottom inset, red arrowheads). (**C**) Microtubules in a given neuronal process span several sections (three shown here; microtubules were traced over 16 sections) and maintain their relative orientations. Microtubules are color coded as in the processes in **B** and were traced and visualized in TrakEM2. (**D**) Synaptic distribution (red, presynaptic site; blue, postsynaptic site) across the arbor of larval neuron A23a. (**E**) Microtubule distribution of larval neuron A23a. Black indicates the microtubule-containing backbone continuous with the soma, green are microtubule-free twigs. See *Video 1* for both microtubules and synapses shown together.

The following figure supplement is available for figure 1:

**Figure supplement 1.** Synapses of neurons with different neurotransmitters.

similar majority of inputs onto twigs in three hemisegmental pairs of premotor interneurons (*Figure 2C*) and brain neurons (*Ohyama et al., 2015*) in the first instar larva (*Figure 2D*). We tested whether the observed distribution of postsynaptic sites onto twigs generalizes across larval stages by comparing a somatosensory interneuron in the first instar to its homologue in late third instar (*Figure 2E*). At both life stages, we find more than 80% of inputs were onto twigs, suggesting that twigs are not a temporary developmental structure. In the adult fly, light microscopy-level analysis of lobula plate tangential cells of the visual system suggests a similar distribution of postsynaptic sites onto twigs (*Leiss et al., 2009*; *Scott et al., 2003*). We annotated EM skeletonizations of medullar Tm3 neurons reconstructed by *Takemura et al. (2013)* in the adult visual system neuropil and found that nearly all their inputs were onto twigs (*Figure 2F*). Our findings suggest that microtubule-free twigs are a general feature of *Drosophila* neurons and constitute the primary anatomical location of synaptic input. Spine-like twigs are found in all cell types, but host a variable, typically non-majority, amount of synaptic input (*Figure 2C–F*). We consider all twigs for the remainder of our analysis.

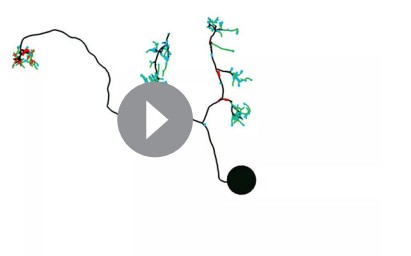

**Video 1.** Rotation of the A23a neuron showing both synapses (red, presynaptic sites; blue, postsynaptic sites) and presence of microtubules (black, with microtubules; green, without microtubules).

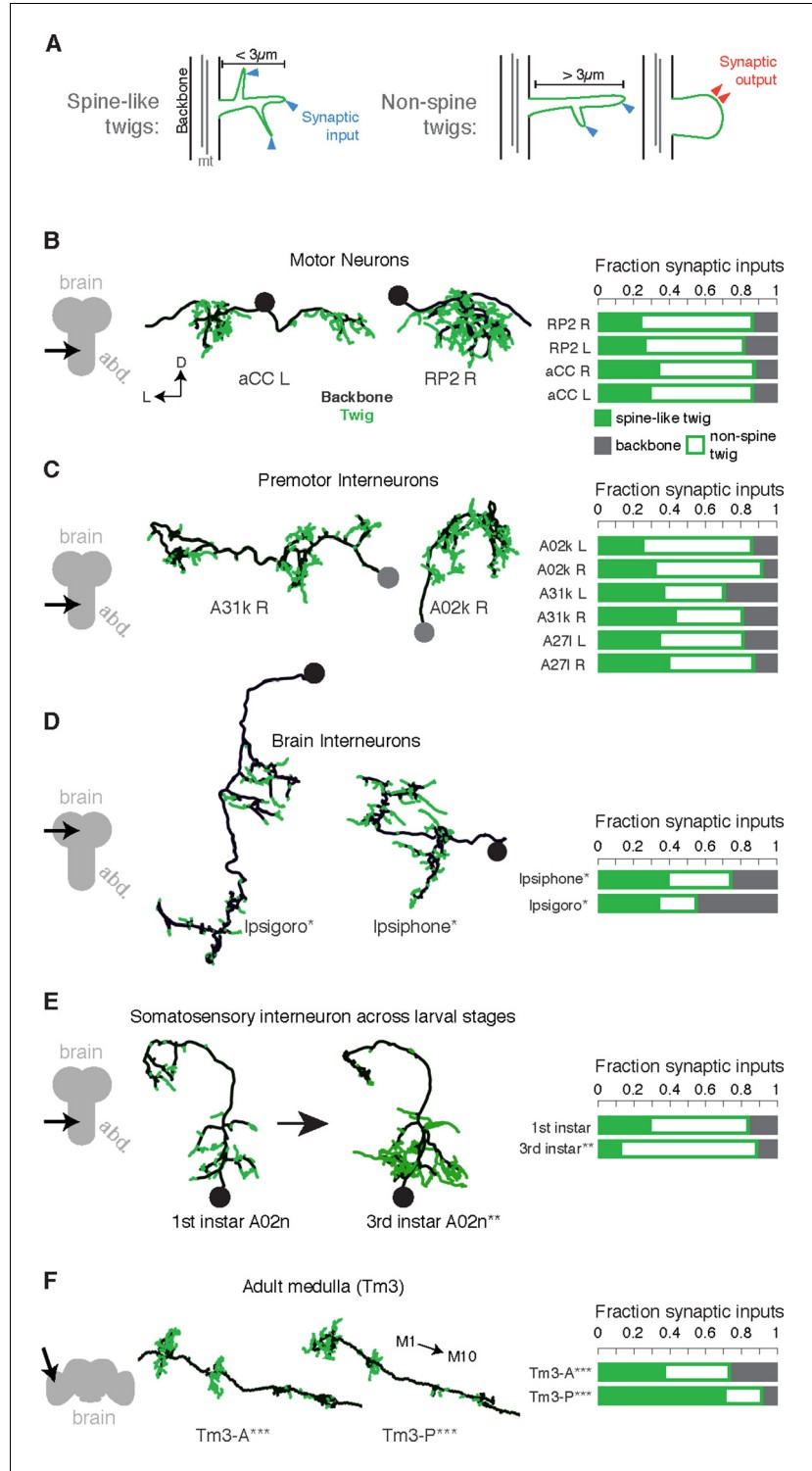

**Figure 2.** Twigs, small microtubule-free neurites, are the primary site of input in *Drosophila* neurons. (**A**) Twigs less than 3 μm are considered spine-like, while those longer or primarily presynaptic are not. (**B–F**) EM reconstructions (middle) of *Drosophila* neurons from different parts of the nervous system (left) showing backbone (black) and twigs (green). At right, the fraction of all synaptic inputs onto short spine-like twigs, longer twigs, and backbone. Data sets are indicated by marks: no asterisk: 1.5 segment volume. *: Whole CNS volume. **: 3rd instar abdominal segment volume. ***: Adult medulla skeletons and images, generously provided by Janelia FlyEM [9]. Neurons are individually scaled to show complete arbors. (**B**) motor neurons in 1st instar larva. (**C**) Premotor interneurons of 1st

*Figure 2 continued on next page*

*Figure 2 continued*

instar larva. (**D**) Interneurons in the brain of the 1st instar larva. (**E**) A somatosensory interneuron cell type across life stages, 1st instar and 3rd instar larvae. (**F**) Tm3 cells in the adult fly medulla.

## Distribution of inputs onto motor neuron dendrites

For a given presynaptic partner, a postsynaptic neuron could concentrate its input synapses onto a single region or distribute them widely. The spatial distribution of synaptic inputs has implications for dendritic processing (*Polsky et al., 2004*), developmental robustness (*Couton et al., 2015*), and as we show, reconstruction accuracy.

To study the relationship between presynaptic neurons and the anatomical locations of post-synaptic sites, we reconstructed all neurons synaptically connected to motor neurons aCC and RP2 in the third abdominal segment of a first instar larva (*Figure 3A–F*).

A dynamically generated and interactive table of synaptic connectivity in CATMAID enabled users to systematically trace all connected arbors. We found 198 identifiable neurons (*Figure 3—figure supplement 1*) and named them according to a developmental lineage-based nomenclature (*Ohyama et al., 2015*), classified 107 other arbors spanning the full segment into eight distinct inter-segmental bundles (*Figure 3—figure supplement 2*), and classified 120 small fragments that could not be joined into larger arbors. We refer to the connection between a pre- and postsynaptic neuron as an 'edge' in the connectivity network, where each edge has a weight equal to the number of synapses between the two neurons. Motor neurons each received between 1 and 28 synaptic inputs from individual presynaptic neurons, with a maximum of 7.3% of all inputs coming from a single neuron (*Figure 3G*). The fraction of synapses accounted for by their presynaptic partners, rank-ordered by number of synapses, is well-fit by an exponential survival function, with a decay indicating that approximately the top 22 presynaptic partners of one motor neuron contribute 63% of all its synaptic inputs (*Figure 3H*).

We next asked how the synaptic input onto aCC and RP2 is distributed across independent twigs. Most individual twigs were small, with the median twig measuring 1 µm in cable and hosting 1 postsynaptic site. The largest typical twig had 16 µm of cable and 20 postsynaptic sites (*Figure 3I*). We find that presynaptic neurons connect to between 0 (backbone only) and 13 twigs, with nearly all connections with 3 or more synapses per edge being distributed across multiple twigs (*Figure 3J*). Similarly, numerically strong edges spanned multiple twigs in the adult visual system Tm3 neurons (*Figure 3—figure supplement 3*).

## Presynaptic sites are associated with mitochondria and microtubules

Different neuronal compartments have different metabolic requirements, such as vesicle recycling at presynaptic sites or restoring resting ion concentrations after postsynaptic response to neurotransmitter signaling (*Attwell and Laughlin, 2001*; *Perkins et al., 2010*). To investigate whether the spatial distribution of mitochondria is a signature of different arbor compartments, we annotated the location of all mitochondria in the four motor neurons and the six premotor interneurons from *Figure 3F* (*Figure 4A–C*). Most mitochondria (348/425) were associated with backbone across motor neurons (*Figure 4D*) and interneurons (*Figure 4E*). Surprisingly, we found that 97% of central presynaptic sites were located within 3 µm of a mitochondrion (*Figure 4F*), although only 47% of cable was located within the same distance. A similar rule did not hold with postsynaptic sites, which were more broadly distributed (*Figure 4G*). This suggests that presynaptic sites and mitochondria are kept near one another, making mitochondrial proximity a useful constraint for validating synapse annotation.

Consistent with this, presynaptic sites were typically also directly associated with microtubules (*Figure 4H*). Approximately 50% of presynaptic sites were located on the backbone and 90% were within 3 µm.

## Circuitry for proprioceptive feedback into a motor circuit

We next looked at the cell and circuit level for regularities that could inform proofreading.

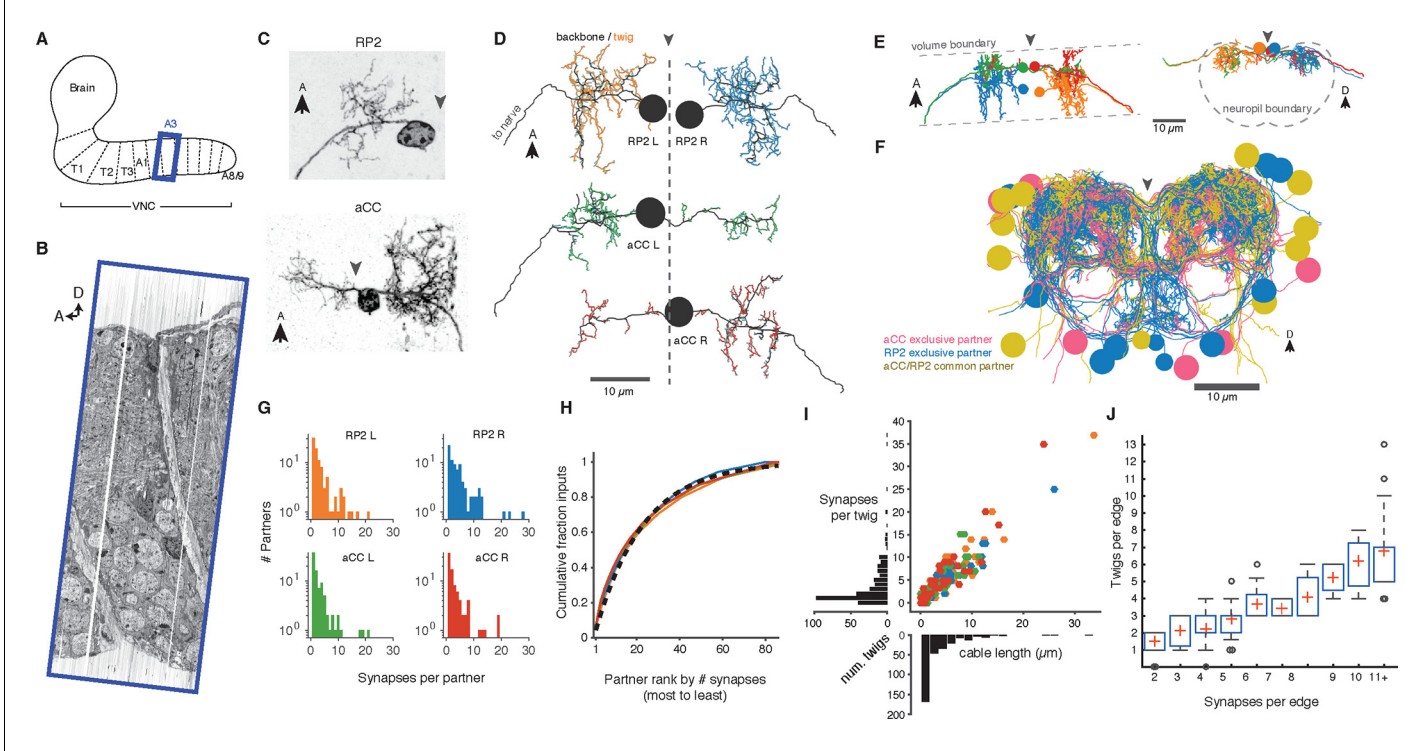

**Figure 3.** Twigs are crucial to larval motor circuitry. (**A**) The EM volume covers one abdominal segment (blue box) of the ventral nerve cord. (**B**) Sagital view of the EM volume. Note segmentally repeated features. (**C**) Dorsal projections of genetically labeled motor neurons RP2 (top, from 1st instar) and aCC (bottom, from 3rd instar). Each cell type has characteristic dendritic arbors. Midline indicated by gray arrowhead. (**D**) EM reconstructions of each of four motor neurons aCC and RP2 in the 1st instar larva match the left and right homologs of aCC and RP2. Backbone is indicated by black, twigs by colors. Midline is shown as dashed line. (**E**) True spatial relationship of the four motor neurons in (**D**), shown dorsally (left) and in cross-section (right). Note that the boundary of the EM volume is limited. (**F**) All arbors presynaptic to aCC and RP2. Colors indicate if neuron is presynaptic to one or both motor neuron cell types. See *Video 2* for rotated views of the arbors. (**G**) Histograms of premotor partners connected via number of synapses. (**H**) Colored lines: the cumulative fraction of total inputs as a function of ranked presynaptic partn ers for each motor neuron are extremely similar. Black dashed line: simultaneous fit for all four motor neurons to 1 - exp (-r/ρ) for rank r gives ρ = 22.34. (**I**) Scatterplot and histogram of the total length and number of synapses on each of the 305 twigs for each of the four motor neurons (colors as previous). (**J**) Number of twigs contacted by motor neuron partners as a function of the number of synapses in the connection. Crosses are median, boxes the interquartile range, whiskers the 10th to 90th percentiles. Outliers shown.

The following figure supplements are available for figure 3:

**Figure supplement 1.** Counts of reconstructed neuronal arbors.

**Figure supplement 2.** Bundles of premotor axons that run the length of the imaged volume.

**Figure supplement 3.** Numerically high synapse edges are distributed over many twigs in adult Tm3 neurons.

In the *Drosophila* larva, developmentally homologous neurons are strongly stereotyped (*Li et al., 2014*), making quantitative analysis of their properties useful for identifying irregularities between homologous cells. Most cell types are represented in the fly nervous system by at least one homologous bilateral pair of individual cells. Bilateral homology suggests that both arbor morphology and synaptic wiring are mirrored, up to developmental noise (*Ohyama et al., 2015*). To let morphology guide proofreading, we developed a collection of neuroanatomical measurements that were independent of absolute location. These metrics, combined with 3d visualization, quickly summarize the structure of complex neurons to help identify and localize inconsistencies (*Figure 5*).

As a case study, we applied our tools to describe a complete sensorimotor circuit. During forward crawling, a peristaltic wave of muscle contraction travels from posterior to anterior segments

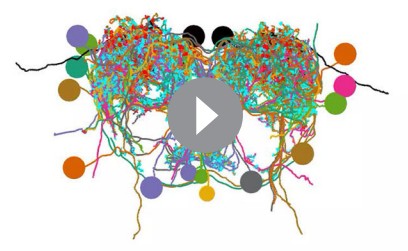

**Video 2.** Rotation of all arbors (colored skeletons) presynaptic to RP2 motor neurons (black skeletons). (Red dots are presynaptic sites, cyan are postsynaptic sites). Dorsal is up.

(*Hughes and Thomas, 2007*; *Heckscher et al., 2012*). Signals from the segmentally repeated proprioceptive neurons dbd have been suggested to act via a polysynaptic inhibitory pathway to stop motor neuron activity after successful contraction of a given segment (*Hughes and Thomas, 2007*). To find pathways between proprioceptive and motor neurons, we further reconstructed axons for proprioceptive sensory neurons dbd, vbd, dmd1, ddaD, and ddaD (*Hughes and Thomas, 2007*; *Grueber et al., 2007*). Because of its implication in proprioceptive feedback (*Hughes and Thomas, 2007*), we further reconstructed all partners of the left and right dbd (*Figure 6A,B*).

Using a graph search within CATMAID, we identified all 1–3 hop pathways from dbd to motor neuron RP2. Comparison of the identifiable intermediate neurons revealed five pairs of homologous neurons with consistent shape, connectivity, and quantitative morphological properties (*Figure 6C,D*). Inconsistencies in any property led to further review to determine if they were due to reconstruction error, true developmental variability (*Figure 6—figure supplement 1*) , or limitations of the raw data. For example, one strong inconsistency in this network, the connection from A02l to A31k (*Figure 6C*), was due to the expected synapse locations being outside the imaged volume on one side but not the other (*Figure 6—figure supplement 2*).

The five pairs of identified neurons could also be matched to light-level images of neurons identified through sparse stochastic labeling (*Nern et al., 2015*) of neurons within a GAL4 expression pattern (*Figure 6D*). Of these, two directly premotor interneurons (A27j and A31k) are immunoreactive to anti-GABA (*Figure 6—figure supplement 3*), whereas the others, all from A02 lineage, are members of the glutamatergic neuron class described in *Kohsaka et al. (2014)*. These novel, putatively inhibitory sensorimotor pathways are well-positioned to mediate a hypothesized 'mission accomplished' signal (*Hughes and Thomas, 2007*). This map also could explain why genetic silencing of A02 neurons was shown to slow peristalsis (*Kohsaka et al., 2014*), as doing so removes a major channel for proprioceptive feedback which is necessary for normal rates of persitaltic waves (*Suster and Bate, 2002*).

## Anatomically enriched wiring diagrams reveal propriomotor circuit motifs

The physiology of synaptic input and output can differ across neuronal compartments. For example, presynaptic inhibition (inhibitory synaptic input onto axon terminals) is important for gain control in fly sensory circuits in a fundamentally distinct manner than dendritic inhibition (*Clarac and Cattaert, 1996*). This suggests that connectivity can be stereotyped at the compartmental level and therefore useful for proofreading. We thus sought a graph representation of a circuit that could faithfully distinguish distinct types of connections (*Figure 7*).

In *Drosophila*, many neuronal cell types have distinct input and output compartments, while a few have entirely intermingled inputs and outputs. Our approach assumes that the neuron can be split into distinct compartments, and at the end checks to see if the split was successful. First, we calculate all paths along the skeleton from each of the neuron's input synapses to each of its output synapses and for each node of the skeleton compute the number of centripetal (toward soma) and centrifugal (away from soma) paths that pass through it (*Figure 7A–B*). This quantity, which we call "synapse flow centrality" (SFC), is analogous to a synapse-specific version of betweenness centrality (*Newman, 2010*). For most neuronal arbors, we find that the most proximal skeleton node with the highest centrifugal SFC corresponds to an intuitive generalization of the locations of spike initiation zones in known polarized neurons of *Drosophila* (*Gouwens and Wilson, 2009*; *Günay et al., 2015*) and other insects (*Gabbiani et al., 2002*).

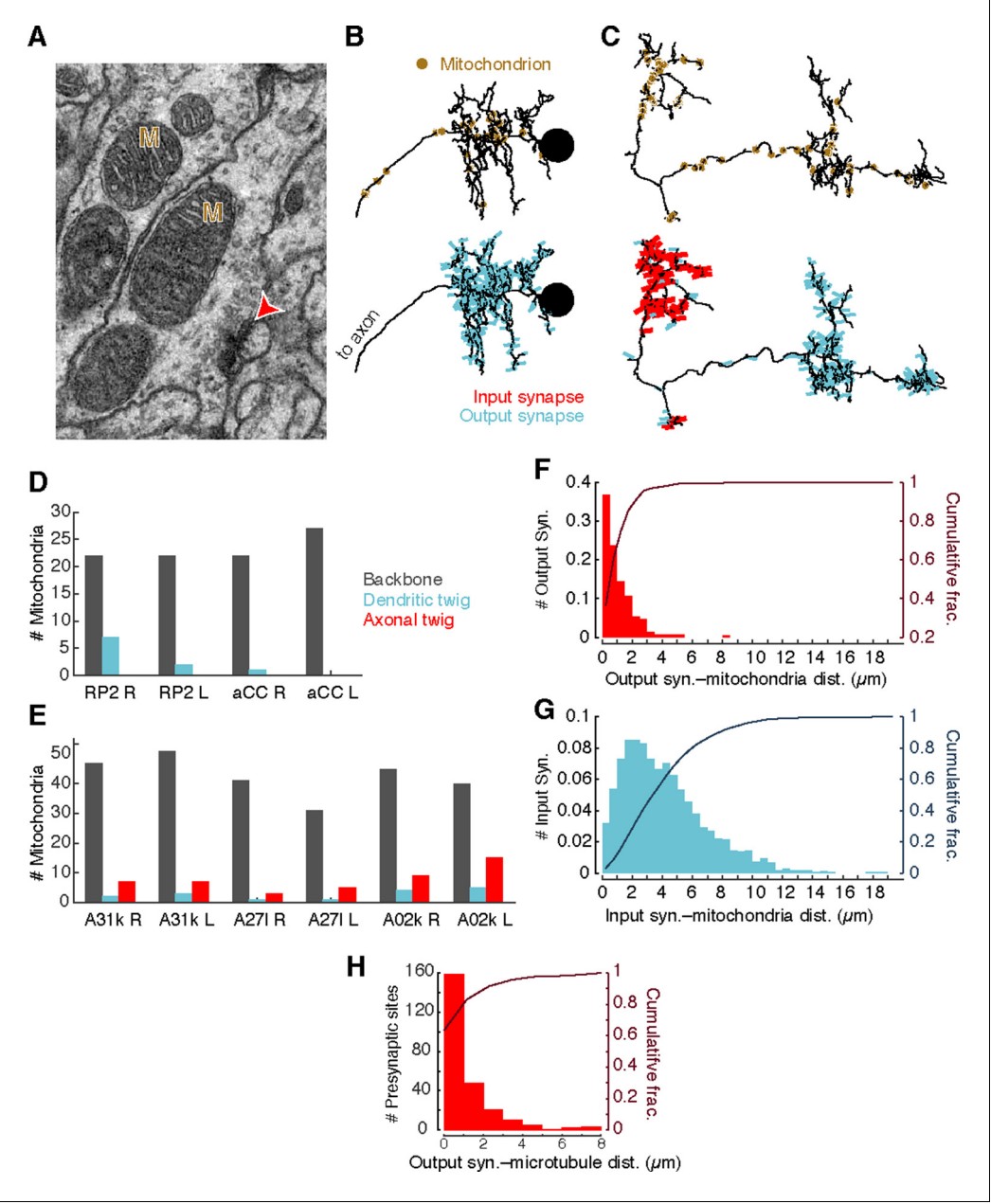

**Figure 4.** Mitochondria are associated with presynaptic sites and cytoskeleton. (A) EM micrograph shows clear mitochondria (labeled with M) and a nearby presynaptic site (red arrowhead). (B) Dorsal view of motor neuron RP2 with locations of mitochondria indicated (top, circles) and synaptic sites (bottom). (C) Dorsal view of interneuron A31k with locations of mitochondria indicated (top, circles) and synaptic sites (bottom). See *Video 3* for both mitochondria and synapses shown together. (D) Number of mitochondria associated with backbone and twig locations on selected motor neurons. (E) Number of mitochondria associated with backbone and twig locations on selected interneurons. (F) Histogram of the distance between presynaptic sites and their nearest mitochondrion along the arbor for the interneurons in E. Cumulative distribution indicated as a line. (H) Histogram of the distance between presynaptic sites and the nearest backbone along the arbor for the interneurons in E. Cumulative distribution indicated as a line.

We quantify how completely input and output are separated on a neuron with a 'segregation index,' an entropy-based measure of the amount of input/output mixing in each compartment, normalized by that of the whole arbor (see 'Materials and methods'). A very low segregation index

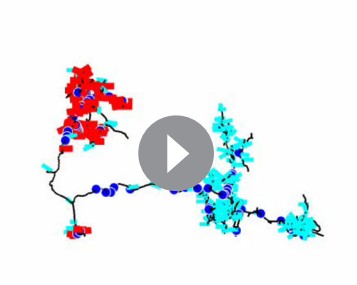

**Video 3.** Rotation of A31k showing both synapses (red, presynaptic sites; cyan, postsynaptic sites) and mitochondria (blue dots). Anterior is up.

means that pre- and post-synaptic sites are inter-mingled and an axon/dendrite compartmentaliza-tion is inappropriate (*Figure 7—figure supplement 1*). Using this approach, we classified all identifiable neurons found in both the left and right hemisegments of the proprio-motor circuitry described above. Of the 3834 synapses between these cells, we found 79% were axo-dendritic (3033), 11% axo-axonic (424), 9% dendro-dendritic (334) and 1% dendro-axonic (43).

We consider two examples of how compart-ment-enriched graphs add important anatomical detail to small microcircuits. First, we analyzed how different proprioceptive inputs converge onto motor neuron RP2 (*Figure 7C–E*). By splitting interneuron A02b into axon and dendrite, we observed that its dendrites receive bilateral proprioceptive input, while its axon synapses both onto the ipsilateral RP2 and axo-axonically onto its strong premotor partner, A03a1 in both hemisegments (*Figure 7C*). In contrast, while dbd only connects indirectly with A02b (*Figure 6C*), it synapses exclusively ipsilaterally and axo-axonicaly onto A03a1 (*Figure 7C*). This suggests that the role of dbd in modulating motor patterns could be qualitatively different than the other proprioceptive sensory neurons, since its direct pathways are typically longer or involve connections types other than axo-dendritic.

Second, we analyzed interactions between the premotor neurons of aCC and RP2 (*Figure 7D,E*). We found that a neuron presynaptic to the aCC motor neuron on both sides targets dendro-dendritically a pre-RP2 neuron (A27h), potentially coordinating the joint excitation of their targets (*Figure 7D*). We also found a premotor interneuron (A27e) with reciprocal connections with a GABAergic premotor interneuron (A27j; see *Figure 6—figure supplement 3*) that receives convergent inputs from dorsal proprioceptive neurons (dmd1, ddaD, ddaE; *Figure 7D*). This suggests that A27j might not only act as an inhibitory premotor input in response to proprioceptive activity, but also have subtler modulatory effects onto other sources of motor input.

Specific connections can also be allocated to specific arbor compartments, which could be used to localize proofreading guided by inconsistencies in connectivity. We thus extended the concept of splitting a neuron into two arbor compartments to an arbitrary number, by defining a compartment as a cluster of synapses near each other along the arbor cable (see 'Materials and methods'). As an example, we consider the axon terminal of dbd, which enters at the interface between two segments and extends symmetric arbors toward the anterior and posterior segments (*Figure 8A*). The synapses form multiple well-separated clusters that we can visualize as a group of graph nodes (*Figure 8B–C*), revealing that the anterior and posterior branches synapse onto homologous interneurons (A08a) for their respective segments (*Figure 8D–E*). This pattern suggests that each A08a cell gets convergent input from the dbd of two consecutive segments, which could reflect that adjacent pairs of segments move together during locomotion (*Heckscher et al., 2012*).

## Proofreading and error correction

Based on the suite of features described above, we developed a two-step iterative method of proofreading after an initial reconstruction (*Figure 9*). An initial systematic review consists in traversing a whole arbor from distal to proximal to follow the expected gradual changes in anatomical properties (e.g. caliber tapering and cytoskeletal changes from microtubule-free to increasing number of microtubules). By freeing mental attention from complex spatial navigation, we found that the systematic review leads to the quick discovery of attentional errors, such as missed synapses, or anatomical inconsistencies, such as a non-contiguous microtubule cytoskeleton. The systematic review status is stored on a per-skeleton node and per-contributor basis (see 'Materials and methods' for details). To allow contributors to incorporate the level of proofreading for each neuron into their evaluation of the neurons and circuits, review status of a neuron is displayed throughout CATMAID, as measured by fraction of the skeleton nodes reviewed. For example, when listing synaptic partners for

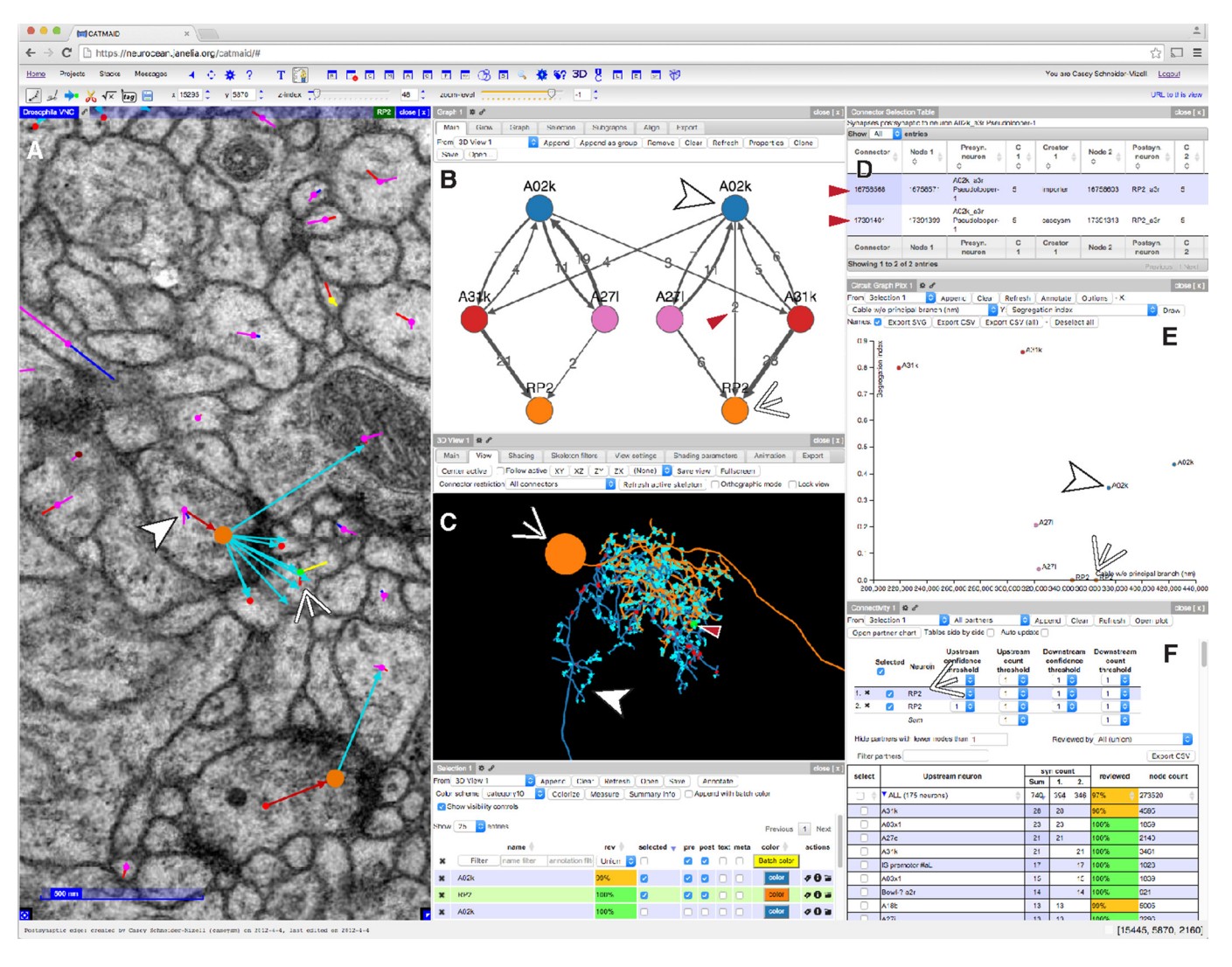

**Figure 5.** CATMAID presents multiple, interactive views on EM imagery and quantitative features. (**A–F**) An example of a CATMAID session in the Chrome web browser (Google, Inc.). Different aspects of a pair of connected neurons, A02k and RP2, are shown across each pane. The number, quantity, location, and neurons in each panel are controllable. (**A**) An image pane shows the EM data, all reconstructed nodes in the view (purple dots), synapse connector nodes (orange dots), and the active node (green dot, indicated by thin white arrowhead). The current active node belongs to an RP2 motor neuron and is postsynaptic to a synapse on interneuron A02k, indicated by the thick white arrowhead. (**B**) Graph representation of a collection of six neurons, including the selected pair indicated as above. Edge labels indicate the number of associated synapse (red arrowhead). (**C**) The pair of neurons indicated in (A), shown in a 3d viewer (orange, RP2; blue, A02k, indicated as above). The active node in the image pane is shown by a green dot in the viewer (indicated by red arrowhead, also the location of the synapse shown at left). (**D**) List of synapses between A02k and RP2, represented in the graph pane by an edge (red arrowhead in B). Each row is clickable, letting the contributor jump to that location to permit fast reviewing of specific connections. (**E**) Plot of quantitative morphological or network measurements of the six neurons in (B). (**F**) Connectivity list shows neurons synaptically connected to selected neurons (here, RP2) and counts the total number of synapses. The row for the presynaptic neuron A02k is offscreen.

neurons of interest, the review status of all partners is shown alongside information such as neuron name and number of associated synapses.

Next, contributors reconstruct the same set of neurons in a different hemisegment (such as the contralateral side) and then inspect high-level quantitative anatomical and connectivity measurements for inconsistencies (*Figure 9*). In most cases, these inconsistencies can be associated with specific compartments of a neuronal arbor, which are then subjected to focused review. This approach helps ensure that the broad structure of the neuron is consistent and that the large branches are correct, as

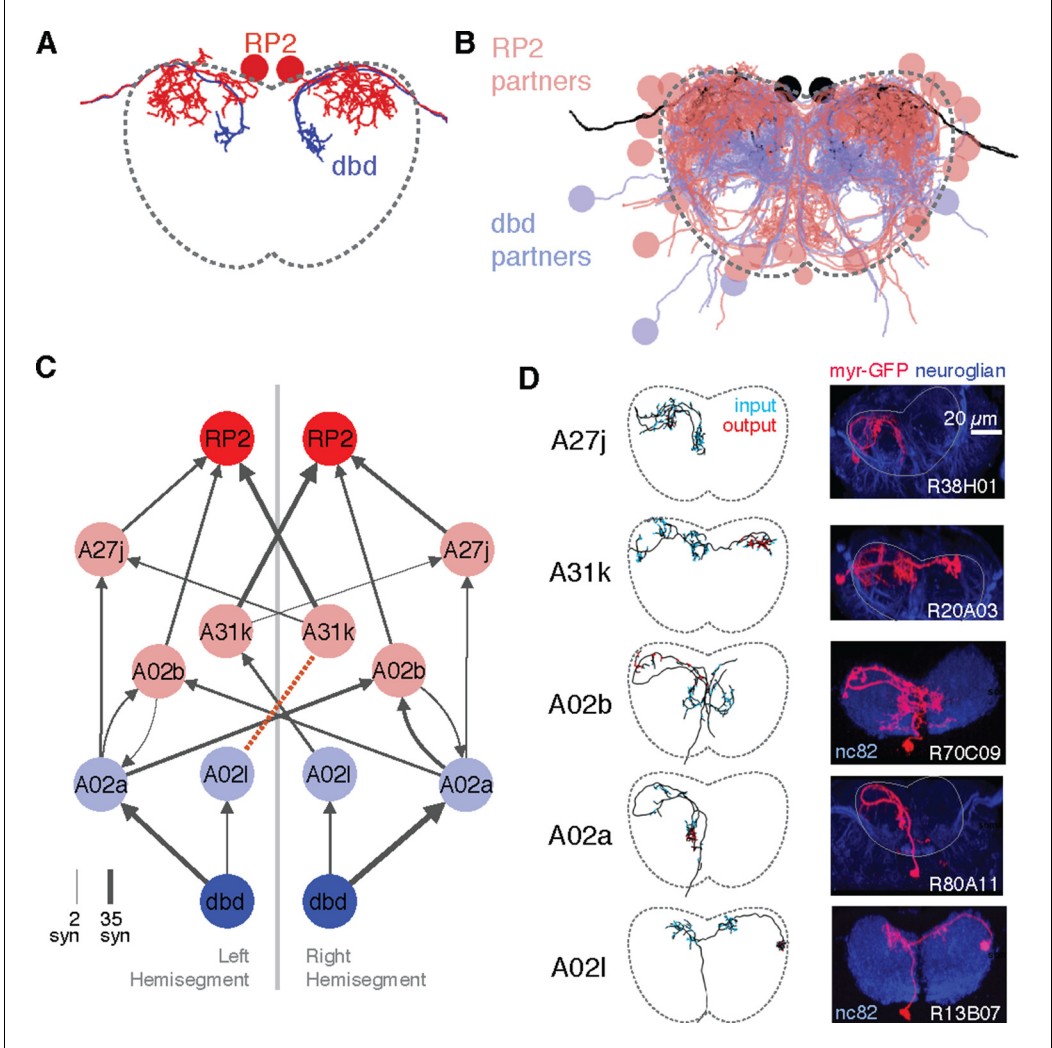

**Figure 6.** Graph search to identify consistent networks. (**A**) The motor neuron RP2 and proprioceptive sensory neuron dbd, shown in transverse. (**B**) All synaptic partners of RP2 and dbd in (**A**). (**C**) Five symmetric pairs of identified neurons link the two cell types with three or fewer hops of at least three synapses each, as found by search in CATMAID. All edges are observed in both the left and right hemisegments, except for a single edge outside the volume boundary (red dashed line , see *Figure 6—figure supplement 1*). Line thickness increases with number of synapses (maximum and minimum values shown). In this and all network diagrams, single synapse edges are not shown for clarity. (**D**) All identified cells in EM (left) could be matched to confocal maximum intensity projections of single neurons found in sparsely labeled GAL4 lines (right, see 'Materials and methods' for details). For neuroglian staining, an approximate neuropile boundary is shown; for nc82 staining, the blue region is a profile of neuropile.

The following figure supplements are available for figure 6:

**Figure supplement 1.** Four pairs of left and right homologs in posterior view, where one of the pairs (canonical) conforms with the arbor shape found in light microscopy (not shown) and the other presents deviations (A,'A': A02d, B,B': A10a, C,C': A23a, D,D': dbd ).

**Figure supplement 2.** Sections were cut approximately 8 degrees from transverse.

**Figure supplement 3.** GABA immunolabeling of proprio-motor interneurons.

errors in them would significantly alter the reconstructed circuit. CATMAID provides tools for transitioning from a potential error identified at a high level, to the images supporting the reconstructed skeletons involved. The key property enabling this is the interactivity of CATMAID's built-in analytical tools that allow for navigating from graphs, 3d views and plots to lists of synapses and spatial locations, and ultimately to the original skeletonizations overlaid on the images (*Figures 5* and *9*).

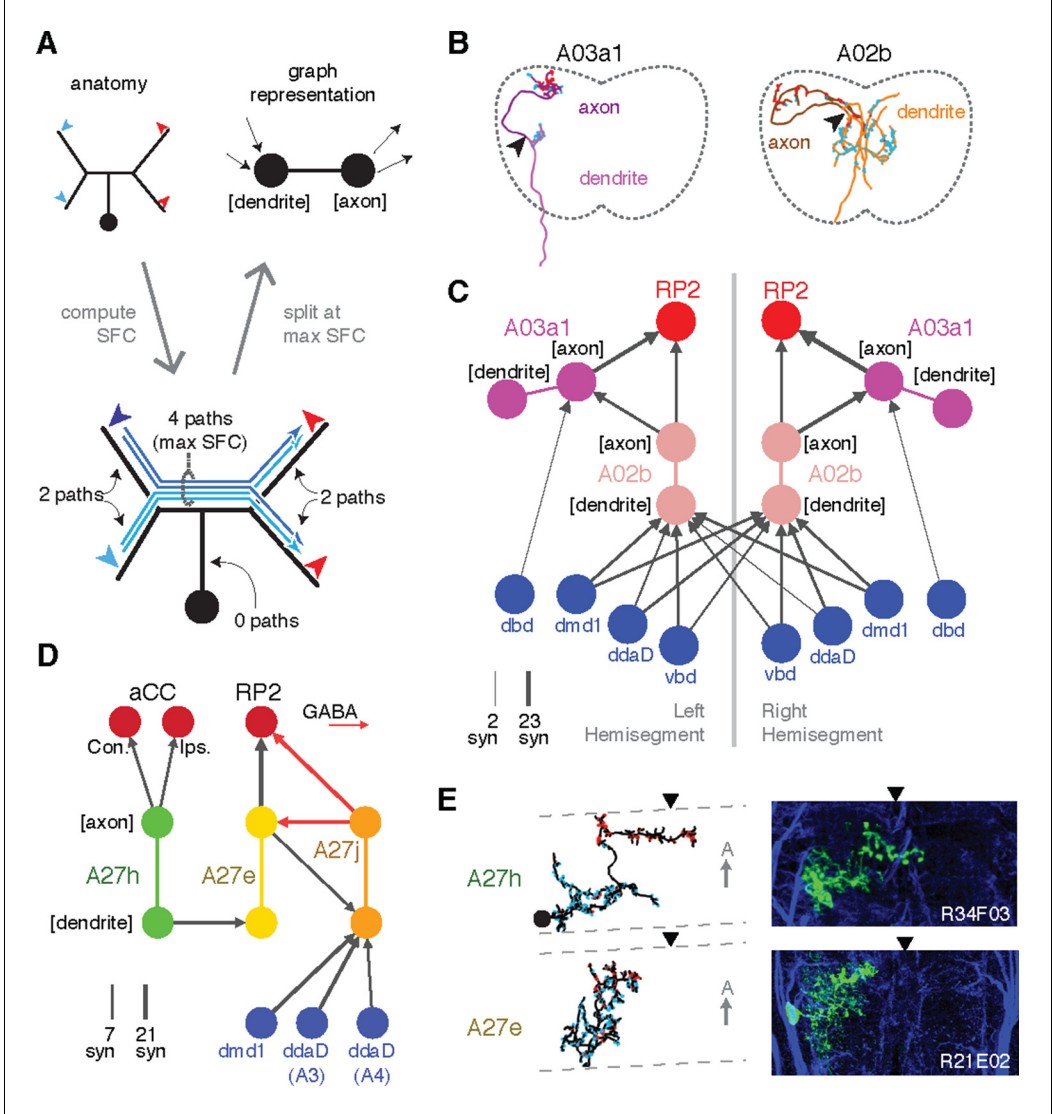

**Figure 7.** Enriching graphs with anatomical compartments. (**A**) Cartoon example of splitting neurons using synapse flow centrality (SFC). (**B**) Examples of two premotor interneurons split into axonal (darker) and dendritic (lighter) regions with this method. Split point is indicated by the arrowhead. See *Video 4* (A03a1) and *Video 5* (A02b) for rotated views of synapses and splits. (**C**) Splitting interneurons into axonal and dendritic compartments in a proprio-motor circuit reveals stereotypic pre- and post-synaptic connectivity to premotor interneuron A03a1 and differential contributions from proprioceptor dbd relative to other proprioceptors dmd1, ddaD, and vbd. Note that the axo-axonic connection from dbd to A03a1 is only 2 synapses, and thus would not appear in *Figure 6A*. (**D**) Splitting interneurons A27j, A27e, and A27h reveals GABAergic pre- and post-synaptic input to a premotor connection, as well as dendro-dendritic coupling between interneurons that connect to synergistic motor neurons aCC and RP2. (**E**) Dorsal projections of A27h and A27e from EM (left) and light (right), as in **C**. Midline indicated by arrowheads.

The following figure supplement is available for figure 7:

**Figure supplement 1.** Neurons are distributed throughout the complete range of possible segregation indices.

Irregularities noticed in higher level features only serve to guide attention, not determine correctness. Any error correction is performed manually on the basis of local information as contained within the EM images (e.g. microtubules, texture, or consistency with neighboring neurites). Despite strong stereotypy in general, developmental variability is present even at the level of high-order branches although often in ways that do not affect connectivity (*Figure 6—figure supplement 1*).

## Validation of the iterative circuit mapping method

Our approach to circuit mapping consists of a single initial arbor reconstruction, followed by edits by the same or different collaborators during proofreading or incidental discovery of errors during subsequent work. Small arbor pieces, left over from pruning when proofreading other neurons or from explorative reconstructions in search of specific neurons, are merged in. We refer to this as "iterative," as compared to consensus methods that combine multiple independent reconstructions (*Helmstaedter et al., 2011*; *Takemura et al., 2013*; *Kim et al., 2014*).

We evaluated the accuracy of our method for *Drosophila* circuits by comparing our results to the those of a consensus approach. We selected six interconnected neurons from the premotor network for independent reconstruction by four individuals. Each individual skeletonized and reviewed his or her reconstructions. Consensus skeletons were then computed for each neuron using RESCOP (*Helmstaedter et al., 2011*).

Both methods resulted in extremely similar arbors, although each method found branches not seen in the other (*Figure 10A*, *Figure 10—figure supplement 1*). All sites of disagreement between the two methods were validated by an expert to determine a gold-standard morphology. Reconstruction and review of these six neurons in the iterative approach took a total of 26.37 hours, while the redundant method by four people took a total of 107.73 hours, almost exactly four times as long.

Existing consensus approaches only calculate neuronal morphology, not synaptic connectivity. Each chunk of the consensus skeleton is associated with the subset of the independent skeletons that are mutually consistent at that location. For some branches, all four individuals agreed, while in others the consensus was based on fewer skeletons. We estimated the connectivity between consensus skeletons by adding each postsynaptic site from each independent skeleton in the consensus, normalized by the number of skeletons contributing to the consensus at that location. Therefore, a given synapse would have a weight of one, the typical value, if it were annotated in all independent skeletons.

We found that both methods recover an identical set of edges in the wiring diagram, with similar number of synapses per edge (*Figure 10B,C*). We next considered the fine differences between consensus skeletons and skeletons reconstructed with our method. The six gold-standard neurons had a total of 1341 postsynaptic sites, with 111 on neurites only present in the consensus skeletons, 229 on neurites only in our method's reconstructions, and 1001 in the arbor found by both. We located 91 missed or incomplete branches (false negatives) in our method, 89 in twigs and 2 in backbones; and 7 incorrect continuations (false positives), 6 in twigs and 1 in backbone. False positives added 30 incorrect postsynaptic inputs in total. Individual missed branches were small in size, complexity, and number of synapses (*Figure 10E–G*), with more than 40 missed or truncated twigs having no influence on connectivity (*Figure 10B,C*). The 3 errors in backbones occurred in small distal dendritic shafts containing one single microtubule, resulting in 7 missed and 4 false postsynaptic sites. Error

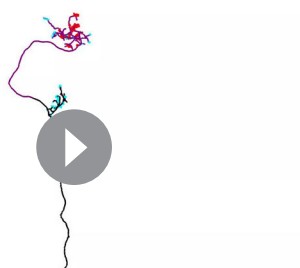

**Video 4.** Rotation of A03a1 showing both synapses (red, presynaptic sites; cyan, postsynaptic sites) and axon/dendrite split (magenta skeleton, axon; black skeleton, dendrite). Dorsal is up.

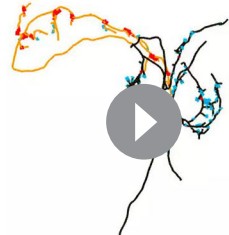

**Video 5.** Rotation of A02b showing both synapses (red, presynaptic sites; cyan, postsynaptic sites) and axon/dendrite split (orange skeleton, axon; black skeleton, dendrite). Dorsal is up.

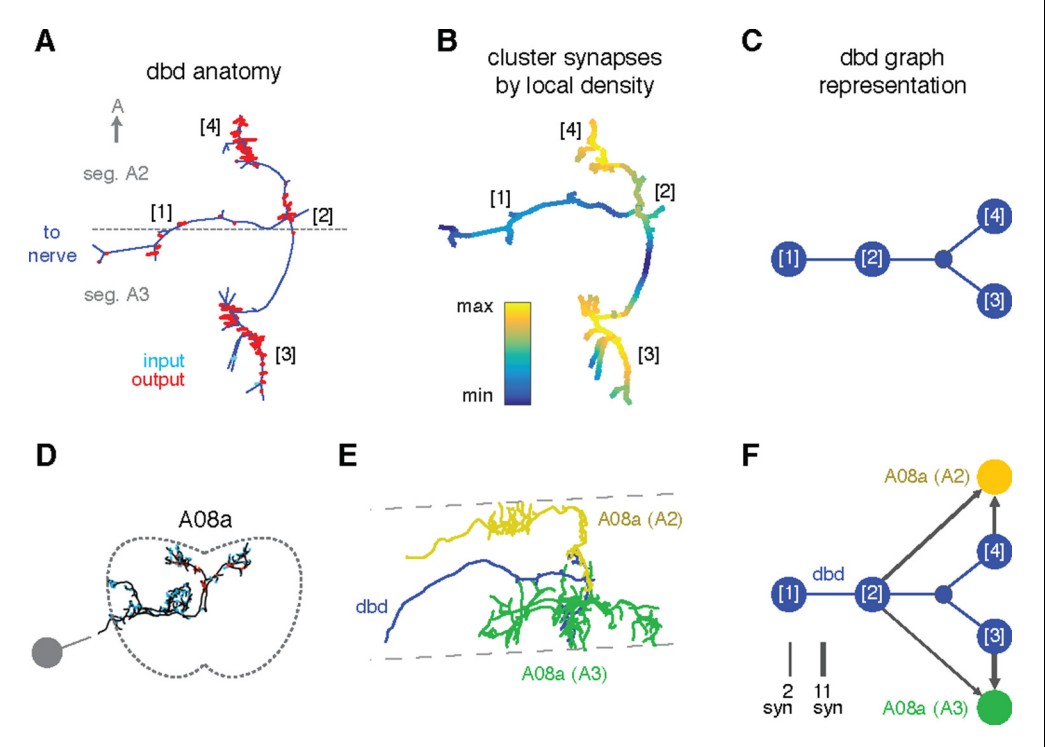

**Figure 8.** Enriching graphs with compartments defined by synaptic density. (**A**) Dorsal view of the axon terminal of dbd. Dashed line indicates, segmental boundary between A2 and A3, numbers indicate clusters of synapses. (**B**) Synapse density mapped onto the arbor. Regions were given a weight as the sum of Gaussian functions ($\sigma = 3\ \mu m$) of the distance to each synaptic site. Colormap is a log scale, arbitrary units. (**C**) Resulting neuron with four nodes, one for the basin of each density peak. Note that the topological structure between clusters (as defined by the peak location) is preserved. (**D**) Transverse view of interneuron A08a, shown here in segment A3. (**E**) Dorsal view of the overlap between dbd (blue) and the A08a in segment A3 (green) and segment A2 (yellow). (**F**) Network of the dbd extended by synapse clustering and A08a. Different clusters have different synaptic regions with the segmentally repeated interneuron.

rates for synaptic output were even lower. The gold-standard neurons had a total of 510 presynaptic sites, of which 509 were found by our iterative reconstructions.

Our data suggest that anatomical structure strongly influences the rate of reconstruction errors in our iterative method. Our total error rate is dominated by false negatives and is much higher for twigs (16.2 µm/error) than for backbone (375.8 µm/error). While attentional errors seemed to dominate missed branches, data ambiguities were often associated with backbone errors. One backbone false merge happened across two adjacent sections in poor registration with one another, while an erroneous truncation occurred across a section where electron-dense precipitate occluded the neurite and its surrounding area.

## Estimating errors in a reconstructed wiring diagram

Neuroanatomy strongly constrains the effect of a typical error on the wiring diagram because, as shown above, the most likely error is to miss a twig and an individual twig hosts few or no synapses.

To estimate the probability of omitting a true edge in the wiring diagram, we analyzed the distribution of synaptic contacts across twigs as a function of the total number of synapses per edge. Edges comprising multiple synaptic contacts were found to be distributed across multiple twigs (*Figure 3J*). With the RESCOP-based validation, we found that our method identified 88% (672/761) of twigs, containing 91.7% of synapses (1230/1341). From these two observations, we estimated the probability of completely missing a true edge as a function of the number of morphological synapses per edge. We found that our method recovers more than 99% of the wiring diagram edges that have at least 3 synapses (*Figure 11A*), assuming twigs are missed uniformly at random (see *Figure 11—figure supplement 1*).

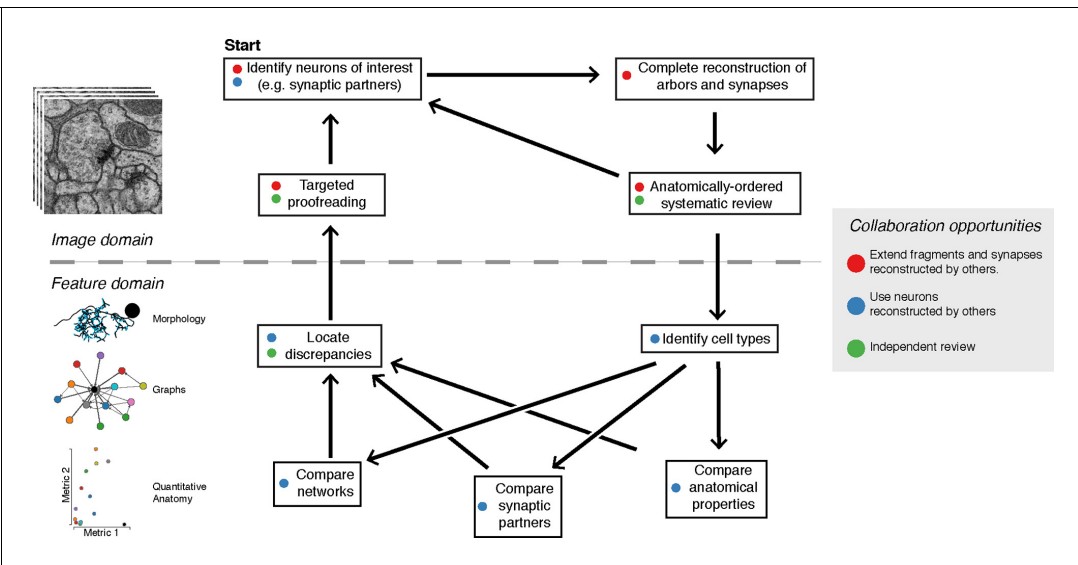

**Figure 9.** The typical reconstruction and proofreading workflow. While reconstruction decisions occur only based on the image data, feature-based comparisons inform specific areas of interest for further proofreading. Each stage in this process can take advantage of the work of other collaborators.

In *Drosophila*, we are primarily interested in the most reliable edges between cell types, as those are most likely to generalize across individual animals. Moreover, we are concerned less about adding extra synapses to true connections and more about adding false edges that would be interpreted as pathways that are not actually present. To estimate the likelihood of introducing a false edge between cell types not just once, but twice (e.g. in a left and right pair of homologs), we simulated adding false twigs to a neuron. The probability of adding a false edge depends both on the probability of adding a false twig (observed false positive error rate: 7 errors in 605 twigs) and the number of nearby but unconnected neurons with presynaptic sites. This will vary depending on the circuit in question. For example, a neuropile with all-to-all connectivity will have no opportunity for false positive edges, while in an array of rigorously separated labeled lines any false positive synapse would be a false positive edge. Further, larger neurons offer more opportunities for false positives than smaller neurons.

For a concrete and realistic example, we consider the motor neuron RP2 (a large neuron). We estimated the number of proximate but unconnected neurons by considering all axons presynaptic to all motor neuron dendritic fields that overlap RP2's dendrites (*Figure 11B*). We assume that a false-positive reconstruction error distributes $m$ synapses across all available axons at random. Even if we assume that $m$ is always among the largest observed ($m = 20$, which is far larger than the average; *Figure 3I*), our model suggests that for the RP2 wiring diagram we can trust symmetric connections of at least 2 synapses (*Figure 11C*). We further note that the small size of individual twigs and the ability in CATMAID to jump directly to the image data associated with synapses comprising an edge make review of a suspect false positive edge extremely fast, on the order of seconds.

Since most errors were of omission and took the form of truncated twigs, we also measured the effect of omitting the distal ends of twigs. Considering again aCC and RP2, we looked at the connectivity observed by considering only synapses located at a given depth into the twig relative to its base on the backbone (*Figure 12A*). With all twigs cropped to zero depth, only inputs onto the backbone remain. More than 90% of postsynaptic sites lay within 5 µm of the backbone (*Figure 12B*). We observed that the first 2 µm already yields at least two synapses, recovering ~90% of the most connected partners. The first 4 µm similarly detects ~90% of all partners with 2 or more synapses and 27/28 pairs of homologous edges (*Figure 12C*). These results indicate that, given the observed distribution of synapses over multiple twigs, wiring diagram edges with many synapses are robust to errors of omission such as truncated twigs. Considering the marginal time involved in reconstructing the full extent of twigs (*Figure 12D*), this robustness could be intentionally exploited towards discovering strong synaptic partners in a time-efficient manner.

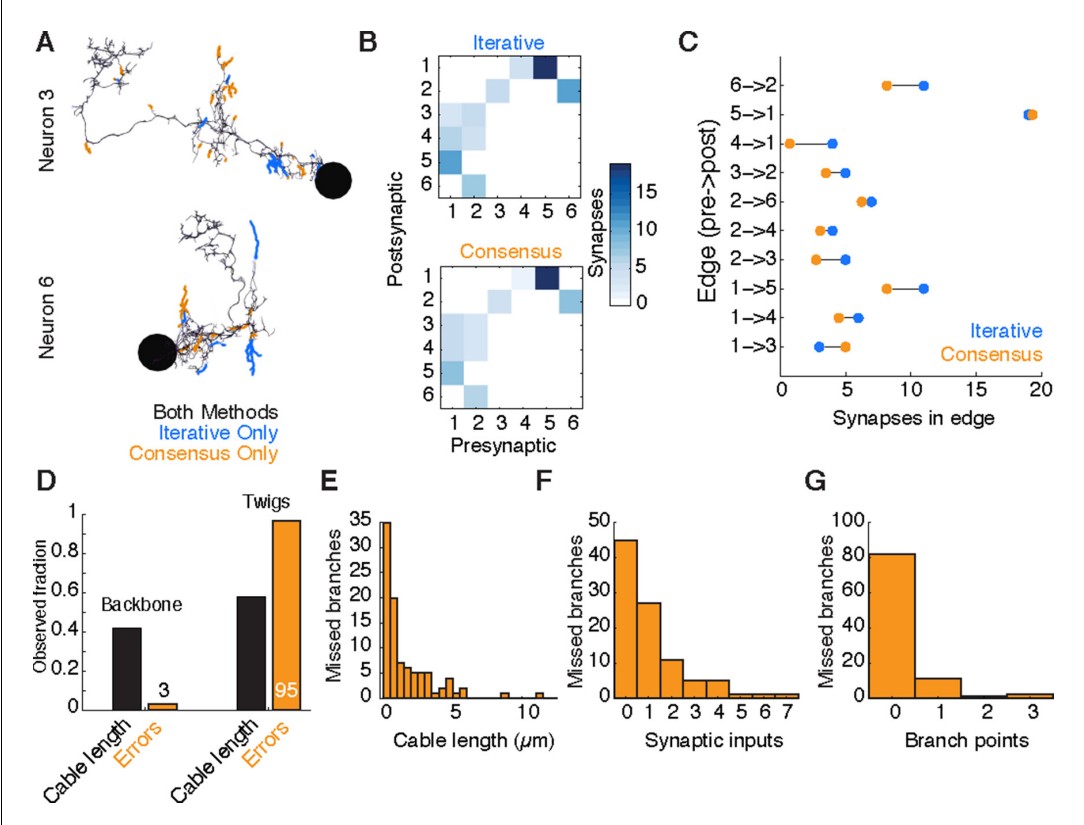

**Figure 10.** Comparison of iterative reconstruction to a consensus method. (**A**) Dorsal view of two of six neurons for which we compared our iterative reconstruction method to a RESCOP-generated consensus of four independent reconstructions. Arbor found in both, dark lines; iterative only, blue; consensus only, orange. (**B**) The adjacency matrix produced by our iterative method has an identical set of edges as that of the consensus method, with variability only in the amount of synapses per edge. (**C**) The weights of each edge (the amount of synapses) are similar between methods. (**D**) Point errors in iterative reconstructions are not distributed equally across the cable of neuronal arbors, instead falling overwhelmingly on twigs. (**E-G**) Branches missed by our iterative method but observed in the consensus method are typically very small and lightly connected as seen from histograms of their (**E**) cable length, (**F**) synaptic inputs, and (**G**) number of branch points.

The following figure supplement is available for figure 10:

**Figure supplement 1.** Four independent reconstructions of a six neuron circuit.

## Discussion

### Neuroanatomy as the foundation for circuit mapping

Neurons are highly structured cells. A human expert's success at circuit mapping from EM volumes stems from the ability to use this structure and apply cell and circuit-level context to interpret nanometer-scale image data. Here, we presented our approach to circuit mapping in EM by building tools in CATMAID that ease and emphasize the use of high level features concurrent with image-level reconstruction. While every reconstruction and editing decision is performed manually, it is informed by a host of quantitative neuroanatomical and connectivity measures computed on demand and, where possible, tightly linked to specific locations in the EM image volumes. In addition to applying existing metrics, we also devised novel algorithms and measures to describe the distribution of synapses across neurons, a feature uniquely well-measurable by EM. Because this method is based extensively on existing information, contributors iterate reconstructions towards more and more correct states. We showed that this more efficiently produces data at least as accurate as computing the consensus of multiple independent reconstructions.

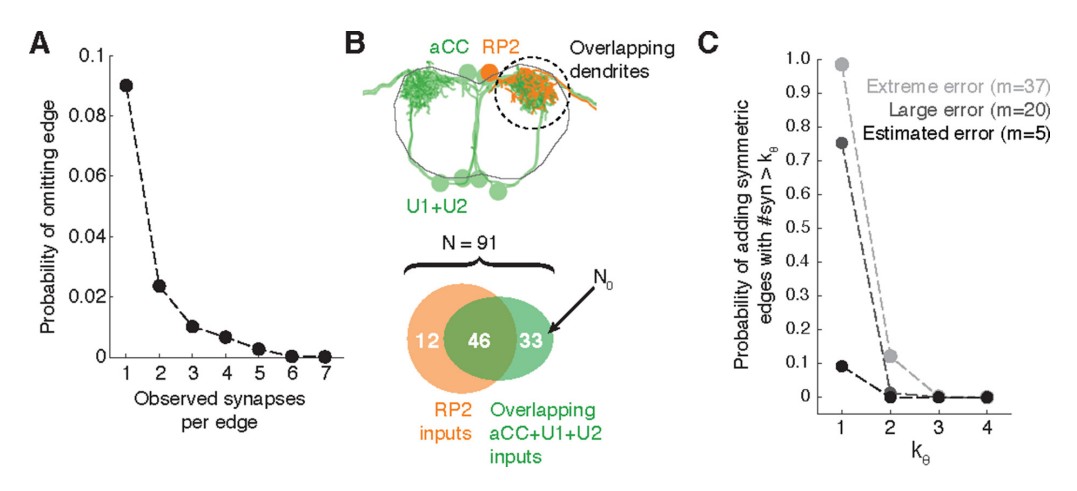

**Figure 11.** Estimating errors that affect graph topology. (**A**) Estimated probability of fully omitting an edge as a function of how many synapses were on the edge based on omitting random twigs with the frequency observed in the validation data. (**B**) Cartoon of dendritic overlap between RP2 and aCC, U1, and U2. On average, 91 axons put at least two synapses on any motor neuron (denoted $N$ in the false positive estimate model, see text for details), of which 33 are not connected to RP2 (denoted $N_0$). (**C**) Probability that, given a pair of homologous postsynaptic neurons, introducing $m$ false inputs randomly distributed across $N$ presynaptic neurons yields at least one pair of false edges of $k_\theta$ or more synapses each. The number of axons were estimated in b, and false input counts are shown estimated from the validation data ($m = 5$), as well as if they came from adding a rare but large twig ($m = 20$), and the largest observed twig ($m = 37$).

The following figure supplement is available for figure 11:

**Figure supplement 1.** We look for clustering in the spatial distribution of errors found by comparison with multiple independent reconstructions.

Central to our approach is the observation that *Drosophila* neurons contain a contiguous microtubule-rich backbone and numerous small microtubule-free distal twigs. We found that small twigs are the primary site of synaptic input for *Drosophila* neurons and that numerically strong connections between neurons are spread across many distinct twigs. If, contrary to observations, neurons were to only contact each other via a single twig that hosts many postsynaptic sites, then this connection would be fragile with respect to developmental noise (*Couton et al., 2015*). Backbones define the spatial extent and stereotyped shape of a neuron, and we found that most presynaptic sites are located on or very near the backbone's microtubules and mitochondria. Our findings are consistent with the notion that metabolic needs and microtubule-based trafficking are limiting factors for the structure of synaptic output.

These different biological requirements for different neuronal compartments are reflected in the rate of reconstruction errors. The large calibers and relatively gradual turns associated with microtubules made errors on backbone less frequent by a factor of nearly 20 relative to on smaller and tortuous twigs. However, we propose that the circuit's resilience to developmental noise, achieved in part by connecting via multiple twigs, underlies the resilience of wiring diagrams to the omission of small dendritic branches, the most typical error observed both here and in reconstructions in the fly visual system (*Takemura et al., 2013*).

Irregularities within a cell type guide review toward small fractions of specific neuronal arbors that could be responsible for a potential error. While reconstructing a neuron, a user can quickly pull up its complete anatomy and connectivity to compare to homologous cells or inspect for irregularities and, crucially, return immediately to the locations in the image data necessary to make the appropriate decisions. We find that this smooth flow from image data to high level features and back to image data—without *post hoc* or offline analysis—is possibly the most important feature in our EM reconstruction technique.

Dispensing with repeated reconstruction without reducing accuracy enables our method to support concurrent neuron reconstruction by many collaborators. This setup prevents duplicated work while ensuring that important locations are visited multiple times. For example, synaptic relations

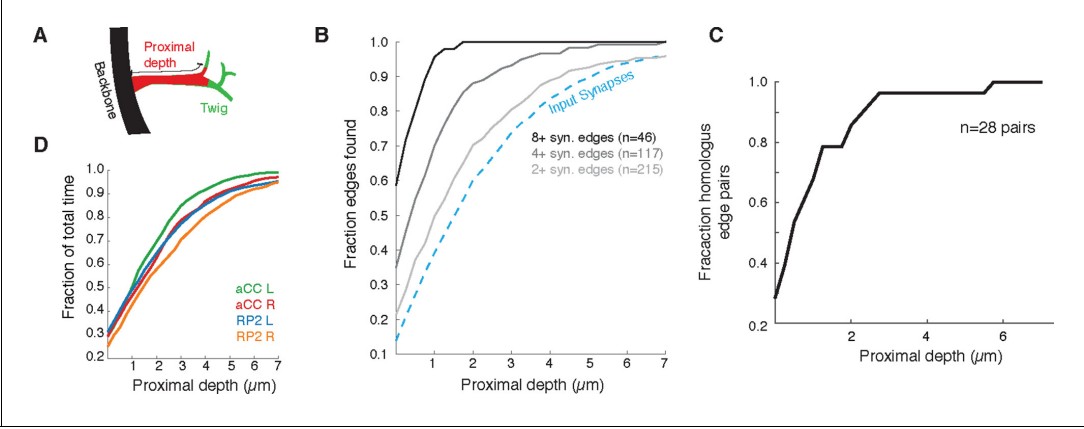

**Figure 12.** Proximal regions of twigs reflect final wiring (**A**) Cartoon of the proximal depth (red) into of a twig (green) measured from from the backbone (black). (**B**) The fraction of two or more synapse edges onto aCC and RP2 that would be found when considering only synapses onto the backbone and twigs cropped at a maximum depth. From light to dark gray are those edges whose final measured connectivity has more than two, four, and eight synapses. Blue dashed line indicates fraction of all input synapses. (**C**) The fraction of pairs of homologous edges from identified neurons (N=28 edge pairs) that would be identified using synapses up to a given depth. (**D**) Fraction of total reconstruction time for each of the four motor neurons (see legend) as a function of cropping twigs at a maximum depth. Note that 0μm depth cropping corresponds to backbone reconstruction only.

are inspected at least twice in different ways, once each from the pre- and postsynaptic side. The presence of existing and correct skeletons in complicated areas, such as registration errors between consecutive sections or gaps, reduces the time necessary for resolving possible ambiguities and effectively provides an extra step of proof-reading by not allowing contradictory reconstructions. Further savings originate in the reuse of data, for example exploratory reconstruction of backbones in search of specific neurons or branches pruned during proofreading are merged into the arbor currently being reconstructed. In summary, in a collaborative environment, the more neurons that are reconstructed, the faster new ones can be added, and the fewer errors existing reconstructions will contain.

Automated methods will be necessary to map circuits with more than a few thousand neurons (*Helmstaedter, 2013*), but they require extensive proof-reading (*Chklovskii et al., 2010*; *Plaza et al., 2012*; *Haehn et al., 2014*). Our methods for analysis of arbor morphology, synaptic distribution and circuit structure and reliability, and their application in proof-reading, apply equally to manually and automatically reconstructed neurons. Neuroanatomical measurements suggest mixed strategies for leveraging both automated algorithms and human effort. For example, mitochondria can be reliably located automatically (*Lucchi et al., 2011*; *Funke et al., 2014*) which, together with our finding of a distance constraint between mitochondria and presynaptic sites, could assist in automated synapse detection (*Kreshuk et al., 2011*; *Becker et al., 2012*; *Kreshuk et al., 2014*). Similarly, the properties of neuronal backbone and twigs suggest that algorithms for the automatic detection of microtubules in serial section EM would be a profitable source of constraints for automated reconstruction of neurites across consecutive sections (*Vazquez-Reina et al., 2011*; *Funke et al., 2012*). While we only considered the relationship between error rate and the presence or absence of microtubules, with the use of automated detection methods it will be important to look at more detailed measures of arbors such as the number of microtubules, curvature, or caliber.

## Generalizability of our iterative reconstruction method to other organisms

Our iterative reconstruction method explicitly uses fundamental properties of dendritic arborizations to achieve circuit reconstruction accuracy without sacrificing speed. Both in insect and in vertebrates, dendritic arbors present two structural compartments: one with microtubules–the shafts–and one without– all the spines and twigs (*Rolls and Jegla, 2015*). Shaft synapses are less likely to be missed because both sides of the synapse present microtubules. Our finding that the connection between

two neurons is resilient to errors of omission of spines or twigs is based on the redundancy afforded by the distribution of postsynaptic sites across the receiving arbor. Instances of this pattern have been observed in mammals in multiple brain regions (*Hamos et al., 1987*; *Bock et al., 2011*; *Kasthuri et al., 2015*). However, the particulars of the distribution of postsynaptic sites over the postsynaptic arbor will be specific to each species—even to brain regions or cell types within each species—and therefore must be measured for the tissue of interest. We have shown that this distribution enables estimating an acceptable false negative rate, the amount of missed synaptic connections that still allow the recovery of strong synaptic connections in the wiring diagram, and thus determines the minimal effort that must be dedicated to twigs or spines.

A fundamental difference between insect and mammalian neurons is that in mammals, the axons can be of small caliber, but represent a substantial fraction of all cable (*Helmstaedter et al., 2013*). Nonetheless, axons need transport of cytosolic elements such as vesicles and mitochondria, which are delivered primarily via microtubules, as well as the anchoring of such elements onto the microtubule cytoskeleton (*Sheng and Cai, 2012*). As we have shown, microtubules provide a strong signal that enables overcoming ambiguities in serial section EM. Therefore cytoskeletal detail, as visible in rich EM images, can greatly facilitate the reconstruction of mammalian axons despite their small calibers.

Catastrophic reconstruction errors are those that affect the backbone of a neuron, dramatically altering the observed circuit wiring diagram. These are generally false positives that originate at ambiguous regions of the image data, and which result in the addition of a large incorrect branch. In *Drosophila*, we exploited the strong morphological stereotypy and unique identity of every neuron to swiftly detect these kind of errors, which are rare, by comparing the overall arbor morphology of homologous neurons across individuals or bilaterally. In nervous systems without uniquely identifiable neurons, these kind of structural errors can be detected either by comparing a reconstructed arbor with prior light-microscopy imaging of the same sample (*Bock et al., 2011*; *Briggman et al., 2011*), or by compiling statistical descriptions of cell type morphology and connectivity (*Sümbül et al., 2014*; *Jonas and Kording, 2015*).

## Sensorimotor circuitry

Larval locomotion, like many motor patterns, results from rhythmic activation of motor neurons (*Heckscher et al., 2012*), but few central components of the underlying premotor circuitry had been identified (*Kohsaka et al., 2014*; *Couton et al., 2015*). Our reconstruction of proprimotor circuitry revealed a complex network comprised of numerous cell types, including a subset of those previously described (*Kohsaka et al., 2014*). We identified a rich collection of local neurons, including neurons anatomically well-suited to provide common drive to synergistic muscles (*Schaefer et al., 2010*) and thus likely a key motor network components. Using anatomically faithful simplifications of neuronal structure, we found several premotor microcircuits employing dendro-dendritic and axo-axonic synapses in parallel with conventional axo-dendritic synaptic connections. For example we found a GABAergic input pre- and post-synaptic to motor neuron input, a motif also observed in mammalian motor circuits (*Fyffe and Light, 1984*).

Although the motor system is rhythmically active in the absence of sensory input (*Suster and Bate, 2002*), proprioceptive sensory feedback is required for natural coordination and timing (*Hughes and Thomas, 2007*; *Song et al., 2007*). We found diverse and complex circuitry for relaying proprioceptive information, including GABAergic and glutamatergic neurons directly relaying proprioceptive input to motor neurons. This motif is well-posed to provide an inhibitory 'mission accomplished' signal to suppress motor neuron activity after a successful contraction during forward locomotion (*Hughes and Thomas, 2007*). However, we also observed that many synaptic partners of dbd were themselves presynaptic to neurons downstream of the other proprioceptive axons, suggesting other, more complex roles for proprioceptive feedback in modulating motor activity. Surprisingly, the axon terminals of proprioceptive neurons themselves almost entirely lacked presynaptic input. This suggests that proprioceptive input is privileged by the larval nervous system and not under fast, dynamic modulation by central circuitry (*Clarac and Cattaert, 1996*), unlike proprioceptive afferents in the locust leg (*Burrows and Matheson, 1994*) and other somatosensory modalities in the larva (*Ohyama et al., 2015*).

Wiring diagrams have been deemed necessary, yet not sufficient, for understanding neural circuits (*Bargmann, 2012*) and a fast approach for discarding hypotheses of circuit function

(*Denk et al., 2012*; *Takemura et al., 2013*). The neuronal wiring reconstructed here offers insights into the structure of an insect motor circuit and its control by sensory feedback, and serves as a complementary resource for detailed functional studies. With the circuit mapping tools and methods demonstrated here, fast, accurate, and targeted reconstruction of circuits in *Drosophila* larva (*Ohyama et al., 2015*) and adult, and other species (e.g. *Platynereis*, *Randel et al., 2015*) is possible.

## Materials and methods

### CATMAID software

We rewrote and greatly developed the Collaborative Annotation Toolkit for Massive Amounts of Image Data, CATMAID (*Saalfeld et al., 2009*) (GPL), to implement our methods for neural circuit reconstruction, visualization and analysis, and with a user and group management system with sophisticated permissions for graded access. The toolkit consists of four parts: (1) the client (a web page), and three types of servers, namely (2) an application server based on the Django web framework (https://www.djangoproject.com), (3) one or more image volume servers, and (4) an instance of the relational database PostgreSQL (http://www.postgresql.org) with all non-image data, which includes metadata such as the spatial information of skeletons, the location of which types of synaptic relations, the text annotations, timestamps and provenance of every action. The original web client accesses, in constant time, arbitrary fields of view of remote stored image volumes. We have greatly extended this capability to include 3-way views (XY, XZ and ZY) and a number of color overlays for multi-channel data such as light-microscopy images or computed derivative data such as membrane probability maps. Analysis of neurons and circuits is performed primarily in the client using the programming language JavaScript, relying on a large number of open source libraries for numerical processing, data management and visualization (D3.js, Numeric Javascript, Cytoscape.js, three.js, jsNetworkX, Raphaël, jQuery, SVGKit). Offline analysis for validation and probability calculations was performed by custom scripts in MATLAB (Mathworks). Documentation and installation instructions are available at http://catmaid.org and code is available at https://github.com/catmaid/CATMAID.

### Serial-section transmission electron microscopy

Wild-type *Drosophila* first instar larval central nervous systems were manually dissected by mechanical separation of the anterior tip of the larva from the posterior portion in PBS, and immediately transferred to 2% glutaraldehyde in 0.1 M Na-cacodylate, pH 7.4 buffer. Samples were post-fixed in 1% $OsO_4$ in the same buffer and stained *en bloc* with 1% aqueous uranyl acetate before subsequent dehydration in ethanol and propylene oxide, and embedding in Epon. Serial 45 nm sections were cut with a Leica UC6 ultramicrotome using a Diatome diamond knife, and picked up on Synaptek slot grids with Pioloform support films. Sections were stained with uranyl acetate followed by Sato's lead (*Sato, 1968*). Sections were then imaged at 4.4 nm × 4.4 nm resolution using Leginon (*Suloway et al., 2005*) to drive an FEI Tecnai 20 transmission electron microscope. The resulting 77,000 image tiles were contrast-corrected, montaged and registered with TrakEM2 (*Cardona et al., 2012*) using the nonlinear elastic method (*Saalfeld et al., 2012*). The generated data volume of 22,775×18,326×462 voxels corresponds to a volume of 91×73×21 $\mu m^3$. The data set covers approximately the posterior half of abdominal segment A2, and a nearly complete abdominal segment A3.

### Preparation of EM images for CATMAID

For display in CATMAID, we Gaussian-smoothed montages of registered EM images (sigma=0.7 pixels, sufficient to remove high-frequency noise to increase the effectiveness of JPEG compression without sacrificing perceptual image quality) and then generated an image pyramid with five zoom levels and diced it to collections of 256 × 256 pixel tiles (512 × 512 and larger can work better for fast Internet connections), stored in JPEG format (75% compression and stripped of headers with jpeg-optim). This approach reduced data storage from over 700 to 90 gigabytes, which were served from a fast seek time solid-state hard drive.

## Server and database configuration

We setup a single server machine (Intel Xeon X5660 with 12 cores, 48 GB of RAM, 10 Gb network card) running Ubuntu 12.04 to host the PostgreSQL database, the image server and the Django server. LDAP id caching was enabled for best performance. Images were stored on high-performance solid-state drives mounted with noatime flag or as read-only, and served via proxy with in-RAM varnishd for caching. The database was configured with large shared buffers (4 GB) and auto-vacuum on (naptime: 8642 min; scale factor 0.4; analyze scale factor 0.2; cost delay -1; cost limit -1) for optimal performance. We chose to serve pages with Nginx, running 8 processes, with epoll on, 768 worker connections, disabled logs and gzip on (except for JPEG image tiles) for best performance, and with public caching and no-expire settings for images. Django was run via Gunicorn with python 2.7 using 8 processes.

## Neuron reconstruction in CATMAID

Reconstruction in CATMAID, as presented here, is based on manual annotation of neurons from image stacks. Where possible, we've honed the interface to reduce the amount of interaction users need to perform while reconstructing neurons. Humans skeletonize a neuronal arbor by placing nodes within a window showing the image stack. A new node, generated by a key press or mouse click, becomes the topological 'child' of an explicitly selected 'active node'. The active node updates to the most recently created node, and branch points are generated by selecting a node that already has child nodes and generating an additional one. When branches are observed, we place a small stub down that branch to ease later follow-up. Since neurons are typically linear at short distances, nodes do not have be manually placed in every image section. In image sections between those containing manually placed skeleton nodes, CATMAID produces intermediate, linearly interpolated virtual nodes that can be interacted with like a real skeleton node. If edited (e.g. moved, attached to a synapse, or used as a branch point), the virtual node becomes a real node manifested in the database.

When a continuation for a neuron has already been reconstructed, the two arbors can be merged with a single click. This is particularly common in the case of synapses, where pre- or post-synaptic placeholder, isolated skeleton nodes (i.e. a skeleton consisting of a single node) are placed to fully describe the synapse. If two arbors both have multiple skeleton nodes, an attempted merge of the two must be confirmed visually in an interactive 3d display, to avoid obvious inconsistencies like multiple soma or an unexpected arbor structure (e.g. spatially intertwined dendritic branches).

Other, optional metadata can also be associated with nodes. A neurite radius can be measured for each skeleton node, with cylinders modeling arbor segments in 3d and spheres modeling the somas. Each skeleton node may also be tagged with any number of arbitrary text snippets to express metadata. Search tools enable finding tags in a specific skeleton or across the data set. Free text can be used to denote structures of interest (e.g. 'Golgi apparatus') or as a personal or team communication convention (e.g. 'check this synapse'). Tags used frequently or to guide analysis (e.g. 'microtubules end') can be mapped to a single key press.

Synapses are made by creating a special class of node (a 'connector') that can be related to a single presynaptic node and multiple postsynaptic nodes. We opt to place the connector node within the presynaptic neuron near the visible characteristics of the synapse, the presynaptic density and nearby vesicle cloud. Each relation is annotated with a confidence value from 1 to 5, with 5 being the default and highest confidence. Again, efficient user interaction simplifies manual annotation. A synaptic connector node can be created with a single click either pre- or post-synaptic to the active skeleton node, based on holding different modifier keys. When the connector node is active, clicking nodes while holding modifier keys produces a pre- or post-synaptic link to it. If no node is already present when the user does this, a new node is generated with the desired link type. Alternative types of connectors with their own constraints (for example abutment or gap junctions) are straightforward to add.

To guide the flow of reconstruction, we rely on special text tags that describe a user's decisions about how to continue a neurite. By default, a node without any child (i.e. a 'leaf node' in the topological tree structure) is expected to need further continuation. When a user decides they have reached the end of a neurite, they can hit the 'K' key to label that node with the text tag 'ends', indicating that no more continuation is necessary. In cases of ambiguity in how to continue, other tags

are used to indicate no further continuation is currently possible: 'uncertain end', to be used if the user cannot decide if a neurite ends, and 'uncertain continuation', to be used if there is an expectation of a continuation due to, for example, microtubules, but the specific continuation is unclear. In effect, then, the process of finishing a neuronal reconstruction becomes a task of continuing every unlabeled leaf node until no more exist. Eventually, all open leaves have been explicitly declared finished and the first draft of the arbor is complete. To ease this process, a single key press ('R') will position the field of view on the closest unlabeled leaf node of a selected neuron, making it simple and efficient to jump to to the next part of the arbor in need of reconstruction.

## Systematic review of a skeleton

Systematic review of a skeleton consists in visualizing each of its skeleton nodes in sequence, adding or editing nodes and synaptic relations as necessary. For this purpose, we partition the arbor to generate the smallest possible set of the largest possible sequences of continuous nodes to minimize the number of times that the reviewer has to switch to a different arbor path. We sort leaf nodes by path length (in number of intermediate nodes) to the root node in descending order. Starting from the most distal leaf node, we generate a sequence of nodes all the way to the root. Then we pick the second most distal node and generate another sequence of nodes until reaching a branch point that has already been assigned to a sequence, and so on for each remaining leaf node. When done, sequences of nodes are sorted by length. The reviewer then iterates each sequence, automatically marking each node as reviewed upon visiting it (using 'q' and 'w' key bindings to go forward and backward in a sequence, and 'e' to jump to the beginning of the longest unreviewed sequence). As a visual aid, each node is centered in the screen, facilitating the detection of changes in the contour of the sectioned neurite, as well as drastic shifts of the field of view that indicate an error (e.g. a jump to an adjacent neurite). The enforcement of a unique directionality and simple one-dimensional path—from distal ends towards inner parts of the arbor or the soma—facilitates noticing glaring inconsistencies such as a path starting off large and microtubule-rich, then reducing to small and microtubule-free, then becoming again large and microtubule-rich. In other words, a review approach coherent with the expected tapering out of neurite caliber and cytoskeleton from soma to distal ends adds context that helps the reviewer.

The total fraction of nodes of a skeleton that have been reviewed is indicated in most tools that can display lists of neurons (e.g. selection table, connectivity tables, connectivity graph), as well as a skeleton coloring mode in the 3d viewer. This enables simple visualization of the current status of review of e.g. all upstream and downstream synaptic partners of one or more neurons in the connectivity widget, of all neurons in a wiring diagram in the connectivity graph, or the review status of a specific branch in the 3d viewer. Given that one or more users may review any node of a skeleton, and the different proficiency of each user, settings allow users to create of a team of other users whose reviews are trusted. Review status visualization can thus be limited to only a user's own reviews, to the union of everyone's reviews, or to the union of all reviews performed by the team of trusted reviewers.

Synapses are effectively reviewed multiple times, given that they are seen from at least two arbors (the pre- and the postsynaptic); more in the case of polyadic synapses, as nearly all synapses in the *Drosophila* larva are. We consider synapses as two elements: the presynaptic relation between a skeleton node and a connector node and the postsynaptic relation between a skeleton node of a neuron and the connector node. Reviewing the associated skeleton node tacitly marks its part of the synapse as reviewed.

To further facilitate systematic review, a 'Skeleton Analytics' tool automatically detects and lists in an interactive table some potential issues that must be addressed in a neuron or collection of neurons. The listing is interactive, allowing jumping to the associated field of view in the image data to determine whether or not the issue describes a genuine error.

- Skeleton nodes tagged with 'ends' which are not leaf nodes.
- Skeleton nodes tagged with 'TODO', 'uncertain end' or 'uncertain continuation'.
- Leaf nodes that are not tagged with neither 'ends' nor 'not a branch.
- Autapses, synapses where the a neuron is pre- and postsynaptic at a single synaptic site. While autapses are known to exist in some nervous systems, all autapses in the larva we have found so far indicated errors in reconstruction.

- Potentially duplicated synapses: when a neuron synapses twice onto another neuron within a small cable distance it is possible that the synaptic active zone has been annotated multiple times erroneously.
- Potentially duplicated postsynaptic relations: when a skeleton receives more than one postsynaptic relation from the same connector, is possible that the extra postsynaptic relations are accidental duplicates rather than, for example, two different dendritic twigs of the same postsynaptic skeleton.
- Lack of a node tagged as 'soma', or the root node of the skeleton not corresponding to a node tagged as 'soma'. The case of multiple somas is generally noticed immediately and addressed without needing special tools.

## Neuroanatomy-driven proofreading techniques

### Arbor structure

A correctly reconstructed neuronal arbor must be biologically plausible. The distribution of microtubules is a biologically grounded approach to subdivide an arbor for analysis. Additionally microtubules are robust to artifacts in serial section EM (e.g. missing sections and noise), since they span many sections while remaining straight and in a consistent configuration within a neurite (*Figure 1*). Errors in the backbone are readily detected by comparing with homologous neurons, or by the extreme and evident consequences of the reconstruction error, such as dramatic changes in direction (sharp angle without branching), the presence of more than one soma, missing large axonal or dendritic trees, or violations of the self-avoidance of neuronal processes, which is often but not always observed.

While most twigs were short and had few synaptic contacts, we did find three interesting outliers (see *Figure 3I*) Each outlier twig was much larger and some included neurites that, upon careful inspection, were in the backbone in their contralateral homolog. This could suggest that parts of the larger twigs could become backbone at a later developmental time point, or they contained labile microtubules that were not captured in the EM sample preparation. Furthermore, these three outlier twigs all had smooth endoplasmatic reticulum at their base and branched very early, suggesting that they are effectively a pair of twin twigs. Considered alone, each half fell within the dimensions of typical large twigs.

### Comparing cell types

Neurons of the same cell type share many properties in common. For *Drosophila*, we define cell type as the pair of left and right homologs, symmetric across the midline, as observed from light-microscopy (*Li et al., 2014*; *Vogelstein et al., 2014*). Most cell types repeat across multiple consecutive segments, though a few do not. Exceptionally a cell type may consist of more than one pair of cells, or of a single unpaired cell with a bilaterally symmetric arbor. Quantitative analysis of the anatomy, synaptic distribution and connectivity for a group of neurons containing potential pairs of homologs helps with detecting potential issues by comparing homologs with each other and with other types.

Reconstructions of homologous neurons can differ due to true developmental differences (*Figure 6— figure supplement 1*), errors in EM reconstruction, misidentification of homology, or asymmetries in imaging data (*Figure 6—figure supplement 2*). To help detect and classify these differences, we generate interactive plots of numerous user-selected metrics on demand. If a pair of homologs is consistently more similar to each other than to other neurons for all three kinds of metrics—anatomy, synaptic distribution and connectivity—the likelihood that the pair contains significant, symmetric, and independent errors is low, and therefore other neurons can be prioritized for review.

We consider several anatomical quantifications that are independent of absolute spatial coordinates and orientation, avoiding issues of mirroring and alignment of neurons (see Suppl. Text). To determine which measures are most helpful for identifying homologous neurons, we applied a distance metric learning method (*Xing et al., 2002*) that scales individual dimensions to minimize Euclidean distance between homologs and maximize distance from other cells. The two most effective measures were total cable length of branches other than the main branch and normalized difference between the number of input and output synapses.

The distribution of synapses over an arbor is characteristic of each cell type. Some cell types have distinct input and output regions with or without dendritic outputs and axonic inputs, while others have arbors with entirely intermingled inputs and outputs (*Figure 7—figure supplement 1*). We devised a novel metric, the 'segregation index', that measures the degree of separation between input and output regions of an arbor. First, we calculate the number of paths from input synapses to output synapses that pass through every node on the skeleton, a quantity we call synapse flow centrality (SFC). We found that the point on the arbor with the highest centrifugal SFC (which considers for any skeleton node only distal outputs and proximal inputs) best separates the axon (distal) from the dendrites (proximal).

To detect and analyze neurons with similar network roles, we reduce neuronal arbors and synapses to nodes and edges in a graph. From the adjacency matrix describing the connectivity, we measure the signal flow and perform graph partitioning via spectral graph analysis (*Varshney et al., 2011*). We find that neurons of the same cell type can group together, even when the number of neurons is variable such as in the optic lobe of *Drosophila* larva (Simon Sprecher, Ivan Larderet & Albert Cardona, unpublished observations).

## Application of prior knowledge to resolve ambiguities

Electron microscopy image volumes of neuropils contain noise. For serial section transmission EM, noise originates during fixation (e.g. broken membranes and reduced extracellular space), serial-sectioning (e.g. folds, cracks, missing sections, thick sections), counter-staining (e.g. precipitated heavy metals, dust particles, or absence of staining due to microscopic air bubbles), and imaging (e.g. locally uneven illumination, tile-wise constant noise originating in improper correction of the camera's dark- and brightfields); for examples see Supplemental Figure 2 in *Saalfeld et al., 2012*.

The most common form of noise consists in missing data either as a partial occlusion of a section, or by the loss of one or more sections. When reconstructing a neuronal arbor, upon reaching an area with missing data (a gap), we use both global and local cues to identify the correct continuation, labeling the skeleton edge that crosses the gap with an appropriate confidence value to express our degree of certainty in the decision. These low-confidence skeleton edges enter into the visualization and analytical tools for further evaluation. Generally, the direction, caliber, and cytoplasmic characteristics of the neuron and its neighboring neurons suffices to identify the corresponding continuation on the other side of the gap. The larger the gap and smaller the neurite, typically the lower the confidence in the identification of the correct continuation.

Locally, gaps up to 500 nm (e.g. 10 serial sections) are crossable using microtubules. The number, direction, and spatial arrangement of microtubules in a neurite are constant over lengths of micrometers, making them reliable structures over many sections (*Figure 1*). Similarly, mitochondria take tubular shapes inside neurites, and their sparseness and relatively constant dimensions identify a neurite across consecutive serial sections (*Figure 1*). Other cues can include the smooth endoplasmatic reticulum that lines large and mid-size neurites; the presence of vesicles of a specific kind (e.g. dark, 50-nm diameter neuropeptide vesicles, or clear large unevenly shaped vesicles, or small, packed clear-core cholinergic vesicles, and others); or other distinctive characteristics such as the presence of microtubules on a specific side of the neurite, or membrane-associated structures, or distinctive cytoplasm texture, such as relative darkness compared to neighboring neurites.

Globally, the properties of a neuronal arbor help to identify continuations across gaps. For example, an axonal neurite tends to continue being axonal in nature within the gap-sized span of a few hundred nanometers; same for dendrites. An obvious feature is that differentiated neurons present a single soma; continuations that lead to a second soma are therefore most likely incorrect.

## A circuit mapping strategy to efficiently identify strongly connected partners

As described above, to identify a neuron quickly in the larva, the first few minutes are best spent skeletonizing the largest structures on the backbone and tracing them to the soma. This minimal representation generally suffices to identify the neuronal lineage and the overall span of the arbor. When the correct neuron has been found, reconstructed in full and reviewed, we begin to map its synaptic partners.

To find out the strongly connected partners of a neuron, we use the connectivity table that aggregates all synaptic relationships, whether with fully reconstructed neurons or single-node skeletons used as placeholders to indicate synaptic partners. Starting at each single-node skeleton, we reconstruct the arbor all the way to the soma by choosing, at every branch point, the larger caliber (may require jumping back to the last branch node occasionally), momentarily ignoring the rest of the arbor. This partial reconstruction suffices to obtain a minimum of information about the partner arbor, such as the lineage. Partner neurons that receive more than one synapse from the neuron of interest will quickly accumulate further fractions of their arbors. These preferred partners—those with many synapses with the arbor of interest—can then be selected for full-arbor reconstruction, while the completion of single-synapse partners (of which a neuron has many, and which in the *Drosophila* larva may play a lesser role in understanding the circuit role of a neuron) can be postponed.

## Dynamically generated and annotated wiring diagrams guide circuit reconstruction and highlight errors

Interactive, partial wiring diagrams calculated on demand during neuron reconstruction guide circuit mapping and the identification of errors. Connectivity-dependent coloring schemes highlight desired features of the circuit, sorting neurons into groups.

A common case is the inspection one or more orders of synaptic neighborhoods. Given one or more neurons of interest (such as RP2), we load all synaptic partners into the graph. For small circuits, visual comparisons between the neighborhoods of left and right homologs can identify similar neurons (e.g. by coloring by stereotyped properties such as the ratio of inputs and outputs, or by their graph centrality; see below) and highlights missing or differently connected neurons, prompting focused review. Coloring the circuit graph relative to a central neuron highlights the relative synaptic order of all other neurons.

Given two neurons, an important circuit question is if there are any paths between them and, if so, through what neurons. This can be queried and added to the graph from within the graph widget, with filters for how many synapses an edge must be.

Other coloring modes include by betweenness centrality (*Brandes, 2001*) of the wiring diagram (calculated as a directed graph), which stresses the role of a neuron within a circuit; and by the percentage of review of the neuronal arbor, indicating at a glance the approximate level of completeness within a group.

When reconstructing neuronal arbors with skeletons, the nodes of the skeletons are annotated with a confidence value signifying the degree of certainty in the continuation of the axon or dendrite. We carry on this confidence into the dynamic wiring diagram representation by splitting the skeleton that models a neuronal arbor at the low-confidence edges, resulting in independent graph nodes. The connectivity of these fragments aids in evaluating their impact on the wiring diagram and their potential correctness.

## Neuron discovery in CATMAID

Neural circuits targeted for reconstruction must be imaged in volumes large enough to encapsulate the complete neuronal arbors of interest (*Helmstaedter, 2013*). Finding specific neurons in unreconstructed data demands prior knowledge, for example using image volumes of genetically labeled neurons, and reference markers like neuroglian or fasciclin II tracts (*Landgraf et al., 2003*) that have anatomical correlates that are conspicuous in EM. The ability to navigate vast volumes at intermediate or low resolution aids in identifying large features such as nuclei, nerves, trachea, or neuropil boundaries, helpful for crossing the resolution gap between light-level microscopy and EM. Although not used in the project described here, to further facilitate finding specific neurons of interest in vast EM volumes, CATMAID supports overlaying other volumetric image data, such as registered confocal stacks.

Given a good guess of the approximate location, finding a neuron of interest involves reconstructing partial backbones (the low-order, microtubule-rich processes). This typically consumes only 10–20 minutes per arbor, due to the large caliber, presence of continuous microtubules, and the paucity of synapses on backbone. In larval *Drosophila*, even partially reconstructed backbones suffice to identify individual neurons by comparing with high resolution single neuron images at light level, given that the lowest-level branches are typically sufficient for unique identifiability of individual

neurons. In our experience, the best starting points are stereotyped features like the main branch points, tracts, commissural crossings, or the neuropil entry point of the primary neurite bundle of all sibling neurons of the same lineage (*Cardona et al., 2010*). Unfinished backbones remain for other contributors to expand or merge into full arbors later, if desired.

## Multi-user editing permissions

In an environment where multiple contributors simultaneously reconstruct neuronal arbors, eventually an ongoing reconstruction reaches that of another contributor. Attempted edits are resolved according to predefined permission rules for who can edit whose work. These rules are implemented as permissions granted to a contributor to alter another contributor's work. The status of "superuser" enables a trusted expert neuroanatomist to edit at will.

Our system operates at two levels: locked and unlocked skeletons. Skeletons that are deemed complete are locked by the contributor, and by default cannot be edited by others unless they have been granted permission to do so. Unlocked skeletons, such as partial reconstructions produced when searching for a specific neuron or when pruning away incorrect branches, can be merged or split by others at will. Neurons are unlocked by default and locking is only to be used upon completion, which prevents sudden and unexpected changes in established results. Individual skeleton nodes and their relations to connectors (which express synapses) can only be edited by the original author, or by others that have been explicitly granted permission to edit the contributions of the original author. In case of conflict or insufficient permissions, a notification system delivers the request to the contributor who can review and effect the change. The result of the work of multiple contributors can be visualized in the 3d viewer, with each node of the skeleton colored according to the identity of the contributor.

## Quantification of the quality of a contributor's work over time

Collaborative reconstructions require that contributors be able to trust the work of others. It is therefore important for a project manager to be able to track the work of each contributor. To estimate an individual's speed and quality, we consider only contributions that have been reviewed by others, within a specific time period. We quantify the number of edits performed by the reviewer, in particular splits (cutting away an incorrect branch), merges (appending a missing branch) and the addition or removal of synapses. While speed and quality are independent, we typically see that better contributors are also faster.

After an initial period, lasting anywhere from a couple of days to about 2 weeks of continuous work, a contributor typically becomes acquainted with the reconstruction task and stops adding erroneous synapses or merging branches from different neurons into one. Remaining errors are typically missing branches or synapses, which are far easier to resolve and have a less significant impact on interpretation of the wiring diagram.

We observe that different areas of the nervous system exhibit profound differences in arbor and synapse morphology, from extensively branching trees in some ventral nerve cord neurons to cloistered self-contacting axons like A02l or in the olfactory lobes (data not shown). Subjectively, contributors that reconstructed neurons in diverse areas of the nervous system experienced a larger variety of shapes and morphologies, which correlated with the acquisition of greater skill.

## Neuron-level annotations

With many expert contributors come many points of view on how to describe neurons. Instead of enforcing a specific ontology, we allow the annotation of any neuron with arbitrary text snippets. These annotations can express a variety of potentially overlapping concepts, from body regions to cell types, gene expression patterns, genetic driver lines and neurotransmitter profiles, among others. The flexibility afforded by the annotation system supports uses from long-term, contributor-centric publication-ready naming schemes to single-use lists helping personal data organization or team collaboration. Our tools allow queries for one or combinations of annotations, as well as metadata such as time or user associated with an annotation.

To make annotations discoverable, we construct a hierarchical tree structure that starts off with three entries: the list of all annotations, the list of all neurons, and the list of all contributors, with each paginated list reducible by regular expression matching. For each annotation, we display five

lists: neurons annotated with it, annotations annotated with it (which act as meta-annotations), annotations that it annotates (acting itself as a meta-annotation), the list of contributors that have used it to annotate an entity (a neuron or an annotation), and the list of co-annotations (other annotations onto the neurons that it annotates). Each annotation, neuron and contributor is expandable, letting the user navigate a graph of relations. For co-annotations, further expansions constrain the listing of neurons to those that share all chosen annotations. For example, starting at annotation 'segment A3', continuing with the co-annotation 'left', and then the co-annotation 'GABA', leads to the listing of all GABAergic neurons on the left hemisegment of abdominal segment A3. Similarly, starting from 'GABA' could lead to 'A3' and 'left' as well, resulting in the same list of neurons. This approach enables the co-existence of many contributor-centric representations of the same neuronal circuits.

Annotations also enable the co-existence of multiple nomenclatures for naming neurons. These could be for example by GAL4 line, by developmental grouping (a name composed of region, segment, lineage and birth order), or by gene expression. In CATMAID, many widgets lists neurons by name. These displayed names are customizable, so that each contributor can see his or her own names, even if the neurons in question were created by others. Each contributor chooses a setting for neuron display names among multiple possibilities, including skeleton IDs, own annotations, all annotations, or most usefully, annotations that are themselves annotated with, for example, 'Janelia GAL4 line' or 'Developmental nomenclature' to indicate naming schemes.

## Synapse clustering algorithm

In order to associate synaptic connectivity not to whole neurons, but to regions of neurons, we adopt an approach where we cluster nearby synapses. Mean shift clustering has been shown to be an effective approach to finding synapse clusters in 3d without assuming a particular number of groups a priori (*Binzegger et al., 2007*). This approach involves convolving synapse locations with a Gaussian kernel to estimate the density of synapses in space. A cluster is then the set of synapses for which, starting at their location, gradient ascent reaches the same density peak. However, locations on one neuron that are close in space can be very far apart along the neuron. Here, instead of considering the density of a neuron's synapses in 3d space, we use a similar procedure to estimate the density of synapses at every point on the arbor (following the cable) and define synapse clusters in the same manner. The only parameter in both approaches is the width of the Gaussian kernel, a physically meaningful parameter.

For these purposes, we define the skeletonization of a neuron to be a graph with a set of nodes $N$ with locations $X_i$ for $i \in N$ and skeleton edges $\varepsilon$ (note that a 'skeleton edge' is between nodes in the skeleton of a single neuron and does relate to synapses). Because the neuron's graph is tree-like, there is a unique non-overlapping path on the graph between any two points $i,j \in N$ with distance $\delta_{ij}$. All synapses (both inputs and outputs) associated with the neuron are represented by the set of their associated nodes, $S \subset N$, noting that the same node can be associated with multiple synapses and thus appear multiple times in $S$. For every node in the neuron graph, we compute the synapse density

$$d(i) = \sum_{j \in S} \exp\left(-\frac{\delta_{ij}^2}{2\lambda^2}\right)$$

where $\lambda$ is a bandwidth parameter that effectively determines the size of clusters, and presynaptic sites of polyadic synapses are counted as many times as they have postsynaptic partners. Note that due to branches, a single synapse close to a branch point may contribute more total density than one that is very distant, reflecting its greater within-graph proximity to more of the arbor. We then look for all maxima in the synapse density and the basins of attraction that flow to them via gradient ascent (i.e. starting at a given node, moving along the maximally positive difference in density between adjacent nodes). A cluster of synapses is then all synapses associated with nodes found within a single basin of attraction of the density function. For neurons found in the 1st instar larva, with $\approx 500$–2000 µm of cable, bandwidths around 8–30 µm provide clusterings that match the subjective description of either 'dendritic arbor' or 'axon'. Smaller bandwidth values result in more granular breakdowns of dendritic and axonal trees (e.g. dbd axons in *Figure 8F*).

## Synapse flow centrality of segments of a neuronal arbor

We define synapse flow centrality (SFC) as the number of possible paths between input synapses and output synapses at each point in the arbor. We compute the flow centrality in three flavors: (1) centrifugal, which counts paths from proximal inputs to distal outputs; (2) centripetal, which counts paths from distal inputs to proximal outputs; and (3) the sum of both.

We use flow centrality for four purposes. First, to split an arbor into axon and dendrite at the maximum centrifugal SFC, which is a preliminary step for computing the segregation index, for expressing all kinds of connectivity edges (e.g. axo-axonic, dendro-dendritic) in the wiring diagram, or for rendering the arbor in 3d with differently colored regions. Second, to quantitatively estimate the cable distance between the axon terminals and dendritic arbor by measuring the amount of cable with the maximum centrifugal SFC value. Third, to measure the cable length of the main dendritic shafts using centripetal SFC, which applies only to insect neurons with at least one output synapse in their dendritic arbor. And fourth, to weigh the color of each skeleton node in a 3d view, providing a characteristic signature of the arbor that enables subjective evaluation of its identity.

## Segregation index: a measure of synaptic sign distribution in a neuronal arbor

A textbook neuron has a purely dendritic arbor and a purely axonal arbor, that is, one neuronal compartment that only receives inputs and another that only delivers outputs onto other neurons. In reality, dendro-dendritic and axo-axonic synapses are present in both invertebrates (*Wilson and Mainen, 2006*; *Olsen and Wilson, 2008*) and vertebrates (*Rudomin and Schmidt, 1999*; *Wilson and Mainen, 2006*; *Pinault et al., 1997*). We have observed that homologous neurons (e.g. identifiable neurons in the left and right hemisegments) have a similar synaptic distribution, which differs from that of other neurons. In *Drosophila*, we find neurons with highly separated input and output (e.g. motor neurons and many types of projection neurons), neurons with entirely intermingled inputs and outputs (possibly non-spiking interneurons [*Burrows, 1992*]), and everything in between.

Having clustered synapses into groups (either by synapse clustering or by splitting the neuron by centrifugal synapse flow centrality), we ask how neuronal inputs and outputs are distributed among the clusters. If the clustering can adequately separate axon from dendrite, then a highly polar neuron will have most of its outputs on the 'axon' cluster and most of its inputs on the 'dendrite' cluster. Motor neurons in the abdominal segments of the *Drosophila* larva are examples of completely polarized neurons. Conversely, highly non-polar neurons can have inputs and outputs distributed homogeneously throughout their arbor. An example of the latter are non-spiking neurons that perform extremely local computations, such as GABAergic antennal lobe interneurons (*Wilson and Laurent, 2005*).

A measure of synaptic sign distribution in a neuronal arbor has the potential to distinguish similar yet uniquely different neurons, as well as to suggest broad functional roles of the neuron. Here, we describe an algorithm to quantify the degree of segregation between input and outputs in a neuronal arbor.

For each synapse cluster $i$ on a neuron with $N_i$ synapses, compute the fraction $p_i$ that are postsynaptic. We measure the uniformity of the distribution of inputs and outputs within cluster $i$ by computing its entropy, for which we consider synapses as entities with two possible states: input or output. At the limits, when all synapses of the cluster are either inputs or outputs, its entropy is zero. When half of the synapses are inputs and the other half are outputs, the entropy is maximal. The contribution of each cluster $i$ to the total entropy is weighted by its fraction of the total synapses (either inputs or outputs).

The entropy of the input/output distribution for each cluster is then

$$S_i = -(p_i \log p_i + (1-p_i)\log(1-p_i)).$$

The total entropy for the arbor is then just

$$S = \frac{1}{\sum_i N_i} \sum_i N_i S_i.$$

However, for reference we need to compare this to an unstructured arbor (i.e. non-clustered) with the same total number of inputs and outputs; for this, we consider the whole arbor as one cluster, and we compute

$$S_{norm} = \frac{\sum_i p_i N_i}{\sum N_i} \log\left(\frac{\sum_i p_i N_i}{\sum N_i}\right) + \left(1 - \frac{\sum_i p_i N_i}{\sum N_i}\right) \log\left(1 - \frac{\sum_i p_i N_i}{\sum N_i}\right)$$

(where $\dfrac{\sum_i p_i N_i}{\sum N_i}$ is just the total fraction of synapses that are inputs).

We define the segregation index as

$$H = 1 - \frac{S}{S_{norm}}$$

so that $H = 0$ corresponds to a totally unsegregated neuron, while $H = 1$ corresponds to a totally segregated neuron. Note that even a modest amount of mixture (e.g. axo-axonic inputs) corresponds to values near $H = 0.5$–$0.6$ (*Figure 7—figure supplement 1*). We consider an unsegregated neuron (H ¡ 0.05) to be purely dendritic due to their anatomical similarity with the dendritic domains of those segregated neurons that have dendritic outputs.

### Reconstruction validation

We validated our iterative reconstruction method, where users' actions are not independent of one another, by comparing our results to that of an established consensus methods involving multiple independent reconstructions. Six neurons (three hemilateral pairs with a total of 2387 μm of cable in the iterative method) were chosen for validation on the criteria that their morphology was well-contained within the EM volume and the entire group formed a connected network. Four contributors (authors CMSM and SG, as well as Ingrid Andrade and Javier Valdés-Alemán) who had little to no prior involvement with the selected neurons were given six seed nodes at the soma of each selected neuron. The number of contributors was chosen based on available, trained users at the time. Each contributor reconstructed the six neurons (skeletons plus synapses) in an otherwise completely unannotated volume that only he or she was working in and then did skeleton-centered review of their own neurons. To determine a consensus skeleton from these four reconstructions, we re-implemented the RESCOP algorithm *Helmstaedter et al. (2011)* in MATLAB (Mathworks, Inc) with slight variations due to differences in the details of skeleton-annotation tools. CATMAID skeletons were resampled so that adjacent nodes were no further than 80 nm apart. Nodes in independent reconstructions were considered consistent if they were within 600 nm of one another, a value chosen because smaller values resulted in correct reconstructions of low-order branch points to be inconsistent.

We developed a minimal method to estimate the consensus connectivity because existing consensus skeleton methods are purely morphological. Any point on the consensus skeleton consists of chunks of one or more skeletons from the individual contributors. We opted to sum all synapses in the consensus skeleton, but to weight each so that if every user annotated the same synapse it would have a total weight of 1. For example, if three contributors reconstructed a given dendritic branch in the consensus skeleton, but only two annotated a postsynaptic site associated with a specific active zone, the consensus synapse would have weight 2/3.

### Model for estimating false negatives and false positives in the wiring diagram

To estimate the probability of completely missing an edge as a function of the number of synapses in the edge, we combine the twig distribution with the error rates obtained from multi-user reconstruction. We found that our reconstruction identified 672 out of 761 twigs, giving our method a recall rate for complete twigs of $q = 0.88$. Let the distribution of $n_b$ twigs across edges with $m$ synapses be $p(n_b; m)$. Assuming each branch has a probability $q$ of being correctly observed, the probability of not observing a specific connection across all $n_b$ twigs is $(1 - q)^{n_b}$. The probability of omitting an edge with $m$ synapses is thus given by

$$P_{loss}(m) = . \sum_{n_b=1}^{m} p(n_b; m)(1-q)^{n_b}$$

In our reconstruction method, we emphasize connections that are found consistently between cells of the same type, typically hemisegmental homologs of a presynaptic and postsynaptic neuron. Using a simple model, we approximate the likelihood of introducing a symmetric, but false, edge between cell types in our wiring diagram due to reconstruction mistakes. Consider two neurons, $j = 1,2$, of the same cell type, with the dendrites of each sufficiently close to $N$ axons to physically permit connections. To add an incorrect edge to the connectivity graph and not just reweight an existing one, any added branches must have synapses from otherwise unconnected neurons. Let the number of axons with zero true connectivity be $N_0$. Assuming symmetry, the number of axons for both neurons should be similar. We then suppose that errors add $m$ synapses to each neuron, with each synapse assigned uniformly at random to axon $i$ G $(1, 2,.., N)$, with the final added edge count onto neuro n $j$ from axon $i$ given by $k_{i,j}$. For clarity, we order the axons such that $i \leq N_0$ designates an axon with no true connectivity. We then ask what is the probability that both $k_{i,1}$, $k_{i,2} > k_\theta$ for any $i \leq N_0$.

The parameters of this model will vary depending on the properties of the neuron and neuropil in question. For example, larger neurons will have more opportunities for error than smaller ones, while neurons with more stringent synaptic specificity have more true zero edges than broadly synaps-ing neurons. To estimate a realistic set of values for the neurons here, we consider the validation data.

Because nearly all false positives occur on the terminal arbors, the number of synapses added by error $m$ can be expressed as $m = rL_t\bar{k}$, the product of the rate of incorrect branches per length of twig $r$, the total length of twigs $L_t$, and the expected synapses per added twig $\bar{k}$. Based on the independent reconstructions, we estimate $r$ as 6 false-positive errors per $1.63 \times 10^3$ μm, $\bar{k} = 5$ synapses, and a typical $L_t = 257$ μm, making $m = 5$. Determining $N$ and $N_0$ is difficult, as it requires knowledge of axons that would not be in the connectivity-driven reconstruction. We estimate reasonable values using the aCC and RP2 network, since aCC dendrites strongly overlap RP2 dendrites, but have several presynaptic neurons not shared with RP2 (*Figure 11B*). In addition to the axons presynaptic to RP2, we find a mean of $N_0 = 36$ inputs exclusive to aCC, so we estimate $N = 87$. We simulated the $10^6$ iterations of the model for $k_\theta = 1\text{-}4$. To investigate more extreme errors than the ones measured, we also considered $m = 37$ synapses, the largest twin twig observed across all neurons looked at, and $m = 20$ synapses, a more typical value for the largest twig of a single neuron.

## Estimating skeleton reconstruction and review time

Skeletons are chimeras, with multiple contributors creating various parts at different points in time. We estimate the total amount of time spent skeletonizing an arbor—including its synapses—by counting the number of 20-second intervals that contain at least one skeleton node or connector node related to the skeleton. This approach is robust to the discontinuity in time and authorship of adjacent skeleton nodes, but tends to overestimate slightly reconstruction time, given the contribution of 20-second intervals for single nodes that were created earlier in time as pre- or postsynaptic placeholder skeletons with a single node, and which were subsequently merged into the growing skeleton. If the latter were each counted as contributing 6 seconds only, reconstruction times per skeleton typically shrink between 15 and 25%.

We estimate the time for the systematic review of a neuron similarly, with the added caveat that parts of the same arbor may have been reviewed more than once. We count the number of minutes for which at least one skeleton node was reviewed, for every contributor that reviewed a fraction of the arbor, and then add up all the minutes of each contributor.

## Larval motor system circuitry

The data volume used was described in *Ohyama et al. (2015)*. It is comprised of 462 sections, each 45 nm thick and imaged at 4x4 nm per pixel resolution. It is bounded anteriorly approximately at the intersegmental nerve entry point in segment A2 and posteriorly near the segmental nerve entry of segment A3. Sections were cut approximately 8° angle relative to true transverse, resulting in a slightly skewed volume with the left side posterior to the right.

Using their characteristic morphology, we identified and reconstructed motor neurons U1, U2, the three VUM motor neurons, aCC, RP5 and RP2 and sensory neurons dbd, dmd1, ddaD, ddaE, and vbd for segment A3. Because dbd, ddaD and ddaE axon terminals also project into anterior and posterior segments, we used segmental repetition to identify the projections of these neurons from adjacent segments that participate in the local circuitry of A3. We chose to focus on dbd, aCC, and RP2 and continued to reconstruct all arbors synaptically connected to the pair of sensory axons and two pairs of motor neuron dendrites for these cells in A3.

We found 425 arbors spanning 51.8 millimeters of cable, with a total of 24,068 presynaptic and 50,927 postsynaptic relations. Nine people contributed data for the finished product: Albert Cardona (200,773/641,740 nodes), Casey Schneider-Mizell (171,718/641,740 nodes), Julie Tran (64,362/641,740 nodes), Stephan Gerhard (34,837/641,740 nodes), John Patton (10,385/641,740 nodes), Ingrid Andrade (8,667/641,740 nodes), Chris Doe (1,505/641,740 nodes), Mark Longair (1022/641,740 nodes), and Akira Fushiki (675/641,740 nodes). Some reconstructions (147,554/641,740 nodes) were imported into CATMAID from prior work in the same volume in TrakEM2 by Albert Cardona, Casey Schneider-Mizell, Mark Longair, Alexander Berthold van der Bourg, and Kenny Floria. All reconstructions were reviewed in CATMAID.

Each arbor was named and described as an identifiable neuron (198 arbors), an ascending or descending projection that spans the full anteroposterior dimension of the imaged volume (107 arbors), a neuron spilling over from adjacent segments beyond the imaged volume (84 arbors), or an unresolvable fragment (36 arbors) (see *Figure 3—figure supplement 1*, *Figure 3—figure supplement 2*). The 198 identifiable neurons amount to 83% of all cable, 88% of all inputs and 62% of all outputs, with ascending or descending projections contributing 29% of all outputs. The anatomy and connectivity of all arbors can be found in the Supplemental Data.

## Immunohistochemistry

CNS of *Drosophila* larva were dissected in PBG (10% NGS [Normal Goat Serum] in 1% PBS) with tweezers under a scope and fixed with 4% paraformaldehyde in 1% PBS for 30 min, washed 3×10 min in PBT (1% Triton-X100 in 1% PBS), blocked for 1 hr in PBG, then washed 3×10 min in PBGT (1% Triton-X100 in PBG), and incubated with primary antibodies (rabbit anti-GABA: Sigma A2053 at 1/1000; chick anti-GFP: Abcam ab13970 at 1/2000) in PBGT for 24 hr at 4°C on small Eppendorf tubes laid on a gentle horizontal shaker. They were then washed 4×15 min in PBT, and incubated with secondary antibodies (goat anti-chick 488: Invitrogen, at 1/500; goat anti-rabbit 568: Invitrogen, at 1/500) in PBGT at 4°C in Eppendorf tubes wrapped in aluminum foil on a horizontal shaker for 24 hr, subsequently washed in PBT 4×15 min (wrapped in foil), and mounted on poly-lysine coated glass slides. Then samples were dehy drated by dipping the slides in an alcohol series (30%, 50%, 70%, 90% in distilled water, then twice 100%) and then in 100% xylene (3 times) using Columbia glass jars with slits for slides; then mounted on glass slides in DePeX (*Li et al., 2014*) using spacer coverslips on the sides. Glass slides were left to dry in a large Petri dish with a lid, wrapped in foil and at 4°C for 3 days. Imaging was done with a Zeiss 710 confocal laser-scanning microscope. Positive immunoreactivity was confirmed by consistent labeling across multiple GFP-labeled cells per imaged nervous system in two or more nervous systems.

## Acknowledgements

We thank James Truman for the use of single cell light-microscopy images; Ingrid Andrade and Javier Valdés-Alemán for reconstructing redundant skeletons for the RESCOP comparison; John Patton, Ingrid Andrade, Kenny Floria, Alex Berthold van der Bourg, Lukas von Ziegler and Julie Tran for reconstructing about 20% of all arbor cable; Daniel Bonnéry for discussions about statistics; Davi Bock for conceiving the notion of low-confidence edge in an arbor; Goran Cerić, Tom Dolafi and Ken Carlile for IT support; Nicholas Strausfeld for the term 'twig' and fruitful discussions; Eric Perlman for code and IT tips; Marta Zlatic, Tomoko Ohyama, Chris Q. Doe, Matthias Landgraf, Akinao Nose, Anna Kreshuck, Fred Hamprecht, Akira Fushiki, Pau Rué, David Wood, Jan Funke, Steve Plaza, Davi Bock and Greg Jefferis for comments; Brett Mensch for discussions. Funding came from the HHMI Janelia Visiting Scientist program (AC), Swiss National Science Foundation grant 31003A 132969 (AC), HHMI, and the Institute of Neuroinformatics of the University of Zurich and ETH Zurich.

## Additional information

### Funding

| Funder | Grant reference number | Author |
|---|---|---|
| Howard Hughes Medical Institute | | Casey M Schneider-Mizell<br>Stephan Gerhard<br>Tom Kazimiers<br>Feng Li<br>Maarten F Zwart<br>Andrew Champion<br>Frank M Midgley<br>Richard D Fetter<br>Stephan Saalfeld<br>Albert Cardona |
| Swiss National Science Foundation | 31003A_132969 | Casey M Schneider-Mizell<br>Stephan Gerhard<br>Mark Longair<br>Albert Cardona |
| Institute for Neuroinformatics, ETH Zurich and University Zurich | | Casey M Schneider-Mizell<br>Stephan Gerhard<br>Mark Longair<br>Albert Cardona |
| International Coordinating Facility (INCF) | Swiss node | Mark Longair |

The funders had no role in study design, data collection and interpretation, or the decision to submit the work for publication.

### Author contributions

CMS-M, SG, Wrote and designed software, Acquisition of data, Drafting or revising the article; ML, SS, Wrote and designed software, Conception and design, Acquisition of data; TK, Wrote software; FL, Acquisition of data; MFZ, RDF, Acquisition of data, Drafting or revising the article; ACh, FMM, Wrote and designed software; ACa, Wrote and designed software, Conception and design, Acquisition of data, Analysis and interpretation of data, Drafting or revising the article

### Author ORCIDs

Casey M Schneider-Mizell, http://orcid.org/0000-0001-9477-3853
Stephan Saalfeld, http://orcid.org/0000-0002-4106-1761

## Additional files

### Supplementary files

• Source code 1. Matlab code for visualizing and computing basic features of the reconstructions, as well as data fully describing the morphology and synaptic connectivity for the proprio-motor reconstructions.

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
