## [Decision Letter]

Thank you for submitting your work entitled "Quantitative neuroanatomy for connectomics in *Drosophila*" for consideration by *eLife*. Your article has been favorably evaluated by Eve Marder (Senior Editor) and three reviewers, one of whom, Ronald L. Calabrese, is a member of our Board of Reviewing Editors, and another is Kristen Harris.

The reviewers have discussed the reviews with one another and the Reviewing Editor has drafted this decision to help you prepare a revised submission.

Summary:

The authors report an enhancement of a semi-automated technique for serial section EM reconstruction of neuronal circuitry in *Drosophila* larva and adults. Specifically, they extended the web-based image data viewer CATMAID (Saalfeld et al., 2009) to enable multiple researchers to map neuronal circuitry. A neuron is reconstructed with a skeleton, a directed tree graph with one or more nodes in every cross-section of neurite in an EM volume. Synapses are annotated as a relation from a node on the presynaptic neuron skeleton to an intermediate "connector node" and then to a node of a postsynaptic neuron skeleton. Reconstructions are immediately synchronized across all users to avoid duplicate or conflicting work, and to take advantage of existing reconstructions to aid further reconstruction and circuit discovery.

They use the method on test cases and make several interesting discoveries that validate the method and contribute to our understanding of *Drosophila* neuronal networks. For example, they show that the vast majority of postsynaptic sites are on microtubule free dendritic twigs, some spine like, and that presynaptic sites are on branches near microtubules and mitochondria. Moreover, they reconstruct a proprioceptive network from sensory axons to motor neurons and provide insightful micro-anatomical comparisons between putative inhibitory and excitatory interneurons.

Essential revisions:

The expert reviews are complimentary and point out two major concerns that should be addressed in revision.

1) The presentation should be redone to clarify the impact of the work and to take full advantage of the *eLife* format. The full impact and import of the technical advance is not made clear to the non-expert reader. A flow chart of the method should be added in the main text (supported by the text) that demonstrates the collaborative process and points out where in the collaborative process each of the 'high-level' metrics/comparisons were made. The interesting biological findings of the paper are not sufficiently explored in the main text but rather relegated to supplementary data. Because there are no figure limitations in *eLife* more of the interesting biological data should be brought forward into the text and perhaps emphasized more in Discussion.

2) There is a concern that the method, while principled effectively for insect neurons, may not be generalizable to other systems. The authors can do more to discuss how their conceptual framework in designing their method can be made applicable in other systems.

The expert reviews are appended because they add specificity to the above concerns and also address more minor points in need of clarification and/or revision.

From Eve Marder, Senior Editor: Please take full advantage of the *eLife* format to make figures as high-quality and large as they need be to make their points. This paper should be visually spectacular, so make it so by taking advantage of figure supplements, or just increasing the number of figures. *eLife* discourages all use of supplemental text and figures, so please bring everything possible into the main body of the text.

Reviewer #2:

Nearly all of my comments reside within the realm of how to make the writing clearer to the reader and are open to the discretion of the authors to implement as suggested, or in different ways.

The data support all of the major conclusions as well as convincingly support the use of an integrative process with its great time-savings, over the consensus model (that requires repeated full reconstructions by many people). When the EM images are outstanding, as they are in this work, then the iterative approach is surely more efficient and equally accurate to a consensus approach.

My one main comment is that the authors need to define the two different approaches in the Abstract, Introduction, Methods, and Discussion more clearly – the references are a little obtuse right now, and even though I think I know the difference (iterative being not repeated full reconstruction of neurons where they are easy to follow vs. consensus being full reconstructions of whole neurons by multiple people – costing 4-10 times the amount of time).

I fully agree from our own experience that curation of difficult points (iteration) in the neuropil by multiple users coming to the synapse from different perspectives, is a much better approach and the data presented here demonstrate it conclusively and should serve to guide future work.

The mitochondrial findings (synapses all within 3 μm of a mitochondrion) have strong functional implications. In of themselves these findings could constitute a whole paper. I would like to see them in a whole paper, or at least brought to the fore in this paper. Burying them in the supplemental data is a disservice to this major finding.

Reviewer #3:

As the authors point out, perhaps the biggest advances in the manuscript are the extensions of CATMAID that enable interactive collaborative analysis of circuits. They make a compelling case for this approach by analyzing networks of tens to hundreds of neurons. However, the manuscript is a complicated mixture of pointing out the unique features of *Drosophila* neurons that make them amenable to the described approach (Figures 1-2, 6) and the description of the actual methods employed and error quantification (Figure 3-5). I understand why the authors organized it as they did, but the necessity to point out the unique features of fly neurons before getting to the various analysis methods raises the question of how generalizable their approach is to neurons in other nervous systems.

Many of the conclusions and error quantifications hinge on the localization of presynaptic input synapses to twigs and the distribution of inputs across twigs. The authors draw the somewhat obvious conclusion that the localization of synapses along neurites will alter the impact of tracing errors. If, however, the morphologies of vertebrate/mammalian neurons are substantially more intricate, it is not clear how many of the principles described in the manuscript will carry over. I am thinking of, for example, thin 100 nm unmyelinated cortical axons extending over millimeters of path length where single errors could significantly impact a connectivity matrix by mis-assignment of all synapses downstream of an error. I commend the authors of making full use of the benefits of *Drosophila* neuroanatomy, but I remain unsure that many of their conclusions are generally applicable to neurons/circuits in other species.

Something like a flowchart demonstrating the collaborative process would also help. I realize this is described in the supplemental text, but I think a diagram would help the reader understand at which point in the collaborative process each of the 'high-level' metrics/comparisons were made. As written the application of each of the analysis tools seems ad hoc. Is this how the authors intend this to be read? This is particularly an issue with Figure 3, which shows 4 different wiring diagrams each using different analysis tools. Given that the authors are not constrained to a 6-figure paper in *eLife* (as far as I am aware), Figure 3 could have been broken out into multiple figures each demonstrating a particular analysis tool in more detail.

---

## [Author Response]

Essential revisions:

The expert reviews are complimentary and point out two major concerns that should be addressed in revision. 1) The presentation should be redone to clarify the impact of the work and to take full advantage of the eLife format. The full impact and import of the technical advance is not made clear to the non-expert reader. A flow chart of the method should be added in the main text (supported by the text) that demonstrates the collaborative process and points out where in the collaborative process each of the 'high-level' metrics/comparisons were made. The interesting biological findings of the paper are not sufficiently explored in the main text but rather relegated to supplementary data. Because there are no figure limitations in eLife more of the interesting biological data should be brought forward into the text and perhaps emphasized more in Discussion.

We appreciate the suggestion to better use the *eLife* format and have made a number of changes to accomplish this. The principle changes to this end include:

A) We added a new Figure 1 with richer synapse and microtubule imagery taken from the previous supplemental figures.

B) We added a new Figure 4 and an associated section in the Results describing the mitochondria data that was previously in a supplemental figure.

C) We added a new Figure 5 showcasing the many views of the same data that our method requires and our software affords. This was based on a previous supplemental figure.

D) We added a new Figure 9 providing a flow chart of the method.

E) We eliminated the supplemental text and moved its contents into the Results and Methods sections.

F) We added several movies that help illustrate the 3d anatomy of neurons shown in main text figures.

2) There is a concern that the method, while principled effectively for insect neurons, may not be generalizable to other systems. The authors can do more to discuss how their conceptual framework in designing their method can be made applicable in other systems.

We have added a Discussion section – “Generalizability of our iterative method to other organisms” – that goes into more detail on the design principles that we believe are universal for neuronal reconstruction. The key points are: 1) measurements of connectivity statistics (e.g. how many synapses are associated with an edge) and the dependence on anatomy of reconstruction errors allow better estimates of acceptable error rates, 2) ultrastructural elements, particularly microtubules, can disambiguate situations where membrane alone is difficult, and 3) statistical properties of morphology and connectivity can help discover outliers that come from false positive errors adding large branches and numerous synapses.

*The expert reviews are appended because they add specificity to the above concerns and also address more minor points in need of clarification and/or revision. From Eve Marder, Senior Editor: Please take full advantage of the eLife format to make figures as high-quality and large as they need be to make their points. This paper should be visually spectacular, so make it so by taking advantage of figure supplements, or just increasing the number of figures. eLife discourages all use of supplemental text and figures, so please bring everything possible into the main body of the text.*

We thank the editor for the encouraging suggestion, and have made the changes described above to accommodate it.

Reviewer #2:

Nearly all of my comments reside within the realm of how to make the writing clearer to the reader and are open to the discretion of the authors to implement as suggested, or in different ways. The data support all of the major conclusions as well as convincingly support the use of an integrative process with its great time-savings, over the consensus model (that requires repeated full reconstructions by many people). When the EM images are outstanding, as they are in this work, then the iterative approach is surely more efficient and equally accurate to a consensus approach. My one main comment is that the authors need to define the two different approaches in the Abstract, Introduction, Methods, and Discussion more clearly – the references are a little obtuse right now, and even though I think I know the difference (iterative being not repeated full reconstruction of neurons where they are easy to follow vs. consensus being full reconstructions of whole neurons by multiple people – costing 4-10 times the amount of time).

We have taken this advice and added a more explicit description of the two methods throughout the paper. In the Abstract, we mention our comparison between the two approaches in the last sentence. In the final paragraph of the Introduction, we clarified the description of iterative versus consensus methods. We have also brought more complete explanations of reconstruction and proofreading into the Methods section.

I fully agree from our own experience that curation of difficult points (iteration) in the neuropil by multiple users coming to the synapse from different perspectives, is a much better approach and the data presented here demonstrate it conclusively and should serve to guide future work. The mitochondrial findings (synapses all within 3 um of a mitochondrion) have strong functional implications. In of themselves these findings could constitute a whole paper. I would like to see them in a whole paper, or at least brought to the fore in this paper. Burying them in the supplemental data is a disservice to this major finding.

We thank the reviewer for the interest in this data, and have moved it to a Results section, “Presynaptic sites are associated with mitochondria and microtubules,” and created a new main text figure (Figure 4) that carries the same data and addresses other issues raised below.

*Reviewer #3:*

As the authors point out, perhaps the biggest advances in the manuscript are the extensions of CATMAID that enable interactive collaborative analysis of circuits. They make a compelling case for this approach by analyzing networks of tens to hundreds of neurons. However, the manuscript is a complicated mixture of pointing out the unique features of Drosophila neurons that make them amenable to the described approach (Figures 1-2, 6) and the description of the actual methods employed and error quantification (Figure 3-5). I understand why the authors organized it as they did, but the necessity to point out the unique features of fly neurons before getting to the various analysis methods raises the question of how generalizable their approach is to neurons in other nervous systems. Many of the conclusions and error quantifications hinge on the localization of presynaptic input synapses to twigs and the distribution of inputs across twigs. The authors draw the somewhat obvious conclusion that the localization of synapses along neurites will alter the impact of tracing errors. If, however, the morphologies of vertebrate/mammalian neurons are substantially more intricate, it is not clear how many of the principles described in the manuscript will carry over. I am thinking of, for example, thin 100 nm unmyelinated cortical axons extending over millimeters of path length where single errors could significantly impact a connectivity matrix by mis-assignment of all synapses downstream of an error. I commend the authors of making full use of the benefits of Drosophila neuroanatomy, but I remain unsure that many of their conclusions are generally applicable to neurons/circuits in other species.

We appreciate this thoughtful criticism, especially as the experience of the authors is dominated by *Drosophila* neurons. First, there remains much fruitful work to be done in stereotyped organisms other than the larva, including the adult fly, *Platynereis*, and *C. elegans*. For such systems, we believe that this method will carry over with only modest changes. For less stereotyped systems such as vertebrates, we believe that several conceptual aspects of our design process are of general use. We have added an extended discussion of how the advances described here would apply in Discussion section “Generalizability of our iterative reconstruction method to other organisms”.

Something like a flowchart demonstrating the collaborative process would also help. I realize this is described in the supplemental text, but I think a diagram would help the reader understand at which point in the collaborative process each of the 'high-level' metrics/comparisons were made. As written the application of each of the analysis tools seems ad hoc. Is this how the authors intend this to be read?

We have added a flow chart (Figure 9) to better outline the method. While all the steps are voluntary to perform, the systematic review status is made clear within individual widgets as described in the Methods section “Proofreading and error correction.” The method described demands that key neurons and connections for a given reconstruction be evaluated by each class of features.

This is particularly an issue with Figure 3, which shows 4 different wiring diagrams each using different analysis tools. Given that the authors are not constrained to a 6-figure paper in eLife (as far as I am aware), Figure 3 could have been broken out into multiple figures each demonstrating a particular analysis tool in more detail.

We have now split Figure 3 into three distinct figures (Figure 6, Figure 7, and Figure 8) and have expanded the clustering figure (Figure 8) to describe the analysis more clearly.